# Low sensitivity of a heavily calcified coccolithophore under increasing $CO_2$: the case study of *Helicosphaera carteri*

**Stefania Bianco**[1,2,4,★]**, Manuela Bordiga**[3,★]**, Gerald Langer**[4]**, Patrizia Ziveri**[4,5]**, Federica Cerino**[3]**, Andrea Di Giulio**[2]**, and Claudia Lupi**[2]

[1]Department of Science, Technology and Society, University School for Advanced Studies IUSS Pavia, Pavia, 27100, Italy
[2]Department of Earth and Environmental Sciences, University of Pavia, Pavia, 27100, Italy
[3]Oceanography Section, National Institute of Oceanography and Applied Geophysics – OGS, Trieste, 34151, Italy
[4]Institute of Environmental Science and Technology, Universitat Autònoma de Barcelona (ICTA-UAB), Barcelona, 08193, Spain
[5]Catalan Institution for Research and Advanced Studies (ICREA), Barcelona, 08010, Spain TS1
★These authors contributed equally to this work.

**Correspondence:** Gerald Langer (gerald.langer@cantab.net)

**Abstract.** Studies on $CO_2$ effects on coccolithophores, unicellular calcifying phytoplankton, show species-specific responses, although only fewer than 5 % of the $\sim 280$ living species have been tested so far. *Helicosphaera carteri* significantly contributes to carbon fluxes and $CaCO_3$ storage due to its size and high calcite production. Despite its importance, few studies have examined *H. carteri* under experimental conditions, and only one has addressed the effects of rising $CO_2$/decreasing pH. *H. carteri* being a large-sized, obligated calcifier species, an important aspect to understand is how changes in seawater carbonate chemistry may affect its morphology. It has already been suggested for other coccolithophores species that the presence of malformed coccoliths may represent a disadvantage for these organisms. Moreover, an alteration in coccolith morphology may affect their contribution to $CaCO_3$ sedimentation and ballasting. As for *H. carteri*, it has also been suggested that due to its high PIC : POC ratio, the species could show a high sensitivity to $CO_2$ rise. In this study, we investigate for the first time whether high $pCO_2$/low pH does affect the morphology of *H. carteri* coccoliths, by culturing this species under pre-industrial $CO_2$ levels ($\sim 295$ µatm) and $\sim 600$ µatm, i.e., the SSP 2-4.5 scenario for 2100 (IPCC, 2021). We also analyzed cellular PIC and POC quotas using morphometric data, roundness, and protoplast and coccosphere size to observe the $pCO_2$ influence on the calcification and photosynthesis ratio.

Our results indicate that *H. carteri* morphology is not significantly affected by increasing $CO_2$, in contrast to other heavily calcified species. The protoplast volume and coccosphere shape of *Helicosphaera carteri* did not vary with changes in $CO_2$, and neither did its particulate inorganic carbon (PIC) and particulate organic carbon (POC) quotas, nor the PIC : POC ratio, indicating unaltered physiological state.

The low PIC : POC ratio found in this work for *H. carteri* compared to ratios previously measured in the same strain under different experimental conditions, and compared to other highly calcified species, could explain the observed low sensitivity of *H. carteri* to $CO_2$. Moreover, the observation of a stable ratio between calcification and photosynthesis in *H. carteri* under increasing $CO_2$ might suggest a constant contribution to the rain ratio under climate change. However, further studies comparing experimental and field data from past ocean acidification events will be required to confirm the conclusions drawn here.

## 1 Introduction

Since the industrial revolution, human activities have led to a rapid increase in atmospheric $CO_2$ concentration. A large amount of this emitted $CO_2$ ($\sim 30$ %) is absorbed by the oceans (Canadell et al., 2007; Sabine et al., 2004), causing a

significant imbalance in the ocean chemistry, which is moving more and more towards lower pH values (IPCC, 2021).

To date, several studies have focused on the effects of seawater carbonate chemistry on calcifying organisms, including coccolithophores (e.g., D'Amario et al., 2020; Dong et al., 2023; Gattuso, 1998; Gazeau et al., 2024; Jokiel et al., 2008; Keul et al., 2013; Langdon et al., 2000; Riebesell et al., 2000; Ries, 2011), and different and sometimes contrasting evidence has been collected for this group (e.g., Iglesias-Rodriguez et al., 2008; Kroeker et al., 2013; Langer et al., 2006; Meyer and Riebesell, 2015; Raven and Crawfurd, 2012; Riebesell et al., 2000). Up to now, these studies have also demonstrated that to predict the responses of this group of calcifying microalgae, the consideration of different species, and even strains (Langer et al., 2009), is required. Indeed, while at the beginning, most of the efforts were focused on common and easy-to-grow species, such as *Emiliania huxleyi* and *Gephyrocapsa oceanica*, in the last 2 decades, many studies have also focused on other species, like *Calcidiscus leptoporus, Calcidiscus quadriperforatus*, *Coccolithus pelagicus* subsp. *braarudii*, and *Scyphosphaera apsteinii* (e.g., Diner et al., 2015; Fiorini et al., 2011; Gafar et al., 2019a, b; Krug et al., 2011; Langer et al., 2006; Langer and Bode, 2011). The latter species are characterized by lower abundances compared to *E. huxleyi* but nevertheless play an important role in coccolithophore $CaCO_3$ production in modern oceans (Baumann et al., 2004; Daniels et al., 2014, 2016; Gafar et al., 2019b; Menschel et al., 2016; Ziveri et al., 2007).

Another low-abundant species but that contributes highly to $CaCO_3$ production is *Helicosphaera carteri*, which is considered one of the main contributors to carbon (C) export and storage into deep-sea sediments (Ziveri et al., 2007), thanks to its large size and higher rates of organic C fixation and calcite production, compared to smaller species (García-Romero et al., 2017; Menschel et al., 2016; Rigual Hernández et al., 2020; Young and Ziveri, 2000). Indeed, while *E. huxleyi* produces between $\sim 6$ and $\sim 20$ pg per cell per day of calcite, *H. carteri* produces between $\sim 80$ and $\sim 120$ pg per cell per day of calcite (De Bodt et al., 2010; Langer et al., 2009; Šupraha et al., 2015). *Helicosphaera carteri* is generally considered to be a species typical of warm waters (e.g., Baumann et al., 2005; Brand, 1994), with moderately high nutrient levels (e.g., Andruleit and Rogalla, 2002; Findlay and Giraudeau, 2000, 2002; Ziveri et al., 1995, 2004). However, it has a general wide distribution (as reported in the CASCADE database; de Vries et al., 2024 TS2), and it seems to be an opportunistic species, easily adaptable to different environmental conditions (Dimiza et al., 2014, and references therein). This adaptability of *H. carteri* is confirmed by its long fossil record, spanning back more than 20 million years (Aubry, 1988; Young, 1998).

Despite its relevant role, only a few studies have been conducted on living *H. carteri* under experimental conditions (e.g., Sheward et al., 2017; Šupraha et al., 2015; Šupraha and Henderiks, 2020), and only one of them considered the effects of $CO_2$ increase on this species (Le Guevel et al., 2024). To assess the potential effects of $CO_2$ increasing and pH lowering on coccolithophores, it is fundamental to study not only changes in calcite production but also in coccolith morphology, as previously suggested by Langer et al. (2011). Both calcite production and coccospheres are beneficial for coccolithophores in terms of eco-physiology and evolution (e.g., Henriksen et al., 2003; Langer et al., 2021; Monteiro et al., 2016; Walker et al., 2018). Coccolith morphology represents a key factor in their ecology (Bown et al., 2004; Young, 1994), and the inhibition or alteration of coccolithophores' ability to calcify can be detrimental for most of the species belonging to this group, as demonstrated by Walker et al. (2018) for *C. braarudii*. Previous studies have also shown that increasing $CO_2$ and decreasing pH can strongly affect coccolithogenesis and coccolith morphology, especially when considering species bearing big-sized and heavily calcified coccoliths, with a possible detrimental influence on the ability of these organisms to face future climate changes (Diner et al., 2015; Kottmeier et al., 2022; Langer et al., 2006; Langer and Bode, 2011).

Given the importance of *H. carteri*'s role in the C cycle, here we investigate for the first time whether rising $pCO_2$ does affect coccolith morphology in this species by analyzing the presence of malformations in *H. carteri* cultures grown under pre-industrial $CO_2$ levels ($\sim 290$ µatm) and $\sim 600$ µatm i.e., scenario SSP 2-4.5 for 2100 (IPCC, 2021). Additionally, we analyze variations in cellular particulate organic (POC), and inorganic (PIC) carbon using morphometric data (e.g., protoplast size, number of coccoliths per coccosphere, coccolith length) and investigate the variations in protoplast and coccosphere size and roundness (RD).

## 2 Materials and methods

### 2.1 Experimental setting and chemical analyses

Monospecific cultures of *Helicosphaera carteri* (strain RCC1323, from the Southern Benguela upwelling area of the South Atlantic from the Roscoff Culture Collection) were grown in natural sterile-filtered seawater collected in the Gulf of Trieste (northern Adriatic Sea, Italy); filtered through 0.22 µm pore size Durapore membrane filters (Millipore); and autoclaved and enriched with vitamins, nutrients, and trace elements following the B medium recipe (CoSMi Trieste, https://cosmi.ogs.it/node/7, Collection of Sea Microorganisms CoSMi Trieste, 2024 TS3). Culture experiments were performed at the National Institute of Oceanography and Applied Geophysics (OGS) in Trieste using the dilute batch culture method (Langer et al., 2013) and keeping constant salinity (35 PSU), temperature (19 °C), light irradiance (100 µmol m$^{-2}$ s$^{-1}$), light / dark cycle (12 : 12 h) under two different levels of $CO_2$ (295 and 600 µatm) in 2.5 L photo-

bioreactors (Kbiotech) controlled by the BioFlex software. A pitched-blade impeller at 100 rpm rotational speed ensured the culture agitation. Before starting the experiments at different $CO_2$ levels, the strain was acclimated for ca. 11 generations to the selected $CO_2$ concentration. Both experiments were run in triplicate. All the experiments were terminated in the exponential phase at low cell density (ca. 10 000 cell mL$^{-1}$), i.e., in dilute batch mode, corresponding to 6 or 7 d from the inoculation.

To calculate the pH values corresponding to the two selected carbon dioxide concentrations, total alkalinity (TA) was measured before starting the experiments. Then, inserting TA, temperature, salinity, phosphate, and silicate data in the CO2SYS program (Lewis and Wallace, 1998), using the constants of Mehrbach et al. (1973) refitted by Dickson and Millero (1987), we identified a pH of 8.18 for 295 µatm and 7.81 for 600 µatm. The pH was maintained at a constant level for the entire duration of the experiments by $CO_2$ injection into the headspace or by adding NaOH (1 M) in the culture through an automated peristaltic pump controlled by the BioFlex software. The pH was measured with a sensor (Hamilton PHI 225; sensitivity 57–59 mV; frequency of measurements 10 s) inserted within the photobioreactor.

To better characterize the carbonate system and the equilibrium among the parameters involved, dissolved inorganic carbon (DIC) and total alkalinity (TA) were measured in two replicas on the final day of the experiment as follows.

For DIC analysis, the culture was filtered through precombusted 0.7 µm nominal pore size glass fiber filters (Whatman GF/F), and two samples of 50 mL were collected minimizing gas exchange with the atmosphere and then poisoned with mercuric chloride (HgCl$_2$) solution in order to prevent biological activity. Samples were stored refrigerated until analyzed. DIC was determined using the Shimadzu TOC-V CSH (combustion system high-sensitivity) analyzer (Shimadzu Corporation, Japan). For DIC, samples were injected into the instrument port and directly acidified with phosphoric acid (25 %). Phosphoric acidification for DIC and combustion conducted at 680 °C, generated $CO_2$ that was carried to a non-dispersive infrared detector (NDIR). The variation coefficient of the analyses was <2 %, and the reproducibility of the method ranged between 1.5 % and 3 %. Typical organic acidification (OA) scenarios do not feature decreasing DIC concentrations. In our experiment the lowest DIC is ca. 1400 µM (high $CO_2$, low pH) and the highest ca. 1700 µM (low $CO_2$, high pH, Table 1). Despite this atypical $CO_2$–DIC combination for OA scenarios, the latter does not undermine the suitability of our experimental setup because DIC is not the parameter of the C system affecting coccolithophores in typical OA studies (Bach et al., 2011; Hoppe et al., 2011; Langer and Bode, 2011). Only under DIC concentrations below ca. 1000 µM, DIC and/or bicarbonate ion concentration might play a role too (Buitenhuis et al., 1999). The parameters of the C system that will have affected *H. carteri* most likely are either pH or $CO_2$ (Bach et al., 2011; Langer and

Bode, 2011); a possible but unlikely candidate is carbonate ion concentration. All three parameters fall within the range of typical OA studies (e.g., Bach et al., 2011; Hoppe et al., 2011; Kottmeier et al., 2022; Johnson et al., 2022; Langer et al., 2009; Langer and Bode, 2011; Milner et al., 2016; Zondervan et al., 2002). Therefore, our experimental setup is suitable for our purpose.

For the TA, 100 mL of culture was filtered through precombusted 0.7 µm nominal pore size glass fiber filters (Whatman GF/F), poisoned with 100 µL of saturated mercuric chloride (HgCl$_2$) to halt the biological activity, and stored in acid-washed borosilicate flasks at 4 °C. TA was measured by potentiometric titration in an open cell (SOP 3b, Dickson et al., 2007) utilizing a nonlinear least-squares approach. The titration was conducted with the Mettler Toledo G20 titration unit interfaced with a computer, using the LabX data-acquisition software. After titration, data were processed, and the TA was calculated using a computer program developed at OGS and adapted to work in association with the Mettler Toledo LabX software and similar to that listed in SOP 3 of DOE (Dickson and Goyet, 1994). The HCl titrant solution (0.1 mol kg$^{-1}$) was prepared in NaCl background to approximate the ionic strength of the samples and was calibrated using certified reference seawater (CRM; Batch no. 107, provided by A.G. Dickson, Scripps Institution of Oceanography, USA). Accuracy and precision of the TA measurements on CRM were determined to be less than $\pm 2.0$ µmol kg$^{-1}$.

The final carbonate system was calculated from temperature, salinity, TA, pH (NBS), phosphate, and silicate, using the CO2SYS program (Lewis and Wallace, 1998), with the same constants mentioned above. The data for the carbonate system are reported in Table 1.

## 2.2 Morphological analyses

*Helicosphaera carteri* coccospheres were collected from triplicate cultures and filtered on cellulose acetate filters (Ø 25 mm pore Ø 0.45 µm) for subsequent analyses at the scanning electron microscope (SEM). Filters were dried at 30 °C for 24 h. The filters were mounted using carbon tapes on SEM stubs and then sputter-coated with gold-palladium using the Emitech K550X/K250 C cathodic metallizer. Analyses at SEM were conducted with a Zeiss Merlin at the Microscopy and X-ray Diffraction Service of the Universitat Autònoma de Barcelona.

After a preliminary observation of the samples, we subdivided the morphologies of *H. carteri* coccoliths in two main categories: normal and malformed (Fig. 1). At the SEM we observed that the malformations occurring in *H. carteri* coccoliths are often characterized by underdevelopment or abnormal development of the flange (Fig. 1c, d). Sometimes, the malformation is also represented by coccoliths presenting a "wavy" shape (Fig. 1c, d). Per sample at least 100 coccoliths were counted, for a total of ∼ 300 coccoliths per experiment (Table 2).

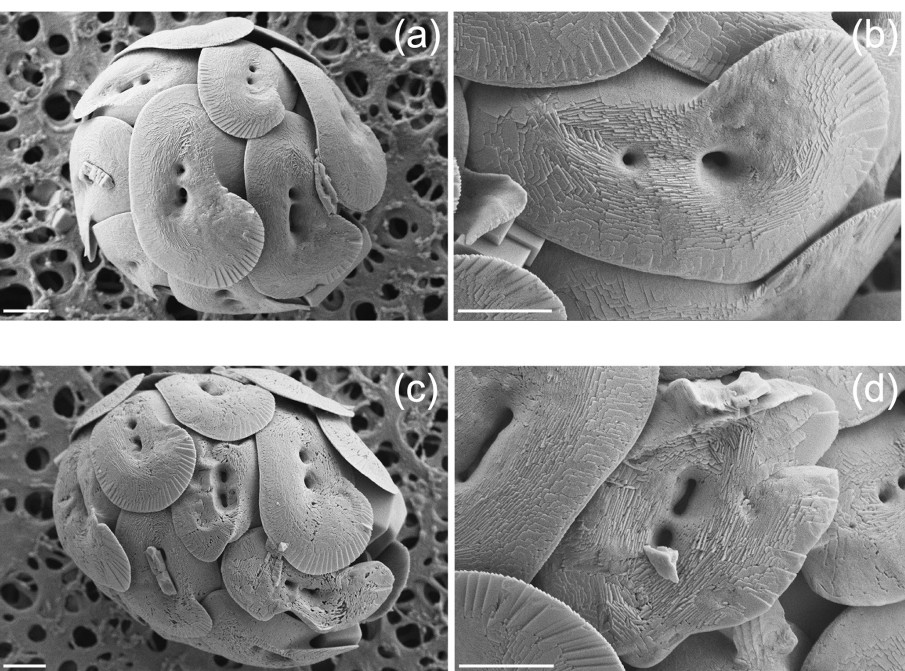

**Figure 1.** Scanning electron micrographs of *Helicosphaera carteri* coccoliths. **(a–b)** Normal and **(c–d)** malformed. Scale bars are 2 μm.

**Table 1.** Parameters of the carbonate system. The values obtained from the CO2SYS program are given in regular font CE1; in bold are the average values directly measured in duplicates per each replica of both the experiments. The average pH values are derived from the whole data collected in continuum along the experiments (pH standard deviation 0.01). SD: standard deviation.

| Parameter | Exp. 295 | Exp. 600 |
|---|---|---|
| $CO_2$ (μatm) | 294.6 | 601.5 |
| SD | 17.84 | 59.74 |
| $CO_2$ ($\mu mol\,kg^{-1}$) | 9.78 | 19.94 |
| SD | 0.59 | 1.98 |
| $HCO_3^-$ ($\mu mol\,kg^{-1}$) | 1413.49 | 1213.70 |
| SD | 106.02 | 144.50 |
| $CO_3^{2-}$ ($\mu mol\,kg^{-1}$) | 141.44 | 51.72 |
| SD | 16.62 | 13.38 |
| **DIC** ($\mu mol\,kg^{-1}$) | **1677.50** | **1374.72** |
| **SD** | **140.87** | **142.03** |
| **TA** ($mmol\,kg^{-1}$) | **1853.82** | **1452.54** |
| **SD** | **166.93** | **146.41** |
| **pH NBS** | **8.18** | **7.81** |
| **SD** | **0.025** | **0.064** |
| $\Omega$ calcite | 3.38 | 1.24 |
| SD | 0.40 | 0.32 |

## 2.3 Morphometric analyses

### 2.3.1 Coccosphere measurements and PIC calculation

On the last day of each experiment, 1 mL of culture was collected and combined with 4 μL of Formalin for coccosphere morphometric analysis. Coccosphere size ($\varnothing$), aspect ratio ($AR_{coccosphere}$), and roundness ($RD_{coccosphere}$) data were obtained by photographing more than 50 coccospheres per each replicate using an inverted microscope Leica CMS-D35578 at $400\times$ magnification and a Leica Camera Ltd CH-9435. The images were processed with ImageJ software (Rueden et al., 2017; Appendix A Fig. A1) using a customized macro (https://github.com/mbordiga/Coccoliths, last access: 27 August 2024). The estimated standard errors of the mean are 0.1219 for $\varnothing$, 0.006119 for $AR_{coccosphere}$ and 0.004549 for $RD_{coccosphere}$ at 295 μatm, while at 600 μatm they are 0.1233 for $\varnothing$, 0.006399 for $AR_{coccosphere}$, and 0.004781 for $RD_{coccosphere}$. Since AR and RD are based on the ratio between major and minor axes of the coccosphere and/or the protoplast, they are considered dimensionless. Hence, the unit for these parameters is not reported.

$AR_{coccosphere}$ and $RD_{coccosphere}$ were strongly correlated ($-0.99$, $p$-value $< 0.0001$); therefore, only RD data have been discussed in this work. RD values closer to 1 indicate a more circular shape (for details see Appendix A Table A1).

The cellular particulate inorganic carbon (cellular PIC) of *H. carteri* was estimated from coccosphere geometry data,

following Young and Ziveri (2000):

$$\frac{\text{PIC[pg]}}{\text{cell}} = C_N \times C_L^3 \times K_s \times \rho \times \left(\frac{M_C}{M_{CaCO_3}}\right), \qquad (1)$$

where $C_N$ is the number of coccoliths per coccosphere, $C_L$ is the coccolith length (µm), $k_s$ is the mean species-specific dimensionless shape factor (0.05 for *H. carteri*; Young and Ziveri, 2000), $\rho$ is the calcite density (2.7 pg µm$^3$), and $\frac{M_C}{M_{CaCO_3}}$ is the molar mass ratio of C and $CaCO_3$ (0.12).

The number of coccoliths per cell ($C_N$) was determined from the samples previously used for counting malformed coccoliths (see Sect. 2.2). At least 50 photographs of coccospheres were captured using the SEM, and the number of coccoliths per cell was estimated by visually counting the visible ones and assuming they represent 75 % of the total (as demonstrated for *E. huxleyi* in Hoffmann et al., 2015).

The average data used for the calculation and the number of individuals analyzed are reported in the Appendix (Appendix A Table A1).

For single coccolith measurements, additional culture samples were obtained by treating 25 mL of culture with 25 mL of a Triton (1 %) and 20 µL bleach solution to separate them from the cell (see Šupraha and Henderiks, 2020).

Part of the obtained pellet was then added to a solution of distilled water buffer with ammonia (1 L distilled water + 30 mL of 25 % ammonia solution). A small amount of this suspension was subsequently pipetted onto a round glass coverslip (∅ 13 mm) and dried on a hot plate at 60 °C. The coverslip was then mounted on SEM stubs (∅ 25 mm) using a carbon disc. To increase the sample's conductivity, four aluminum bridges connecting the coverslip to the edge of the stubs were added in each sample. The samples were then sputter-coated with platinum and analyzed using the Tescan Mira3XMU SEM of the Department of Earth and environmental Sciences at the University of Pavia (CISRiC-Arvedi Laboratory). Unfortunately, due to an alteration in the preservation state of the material, it was not possible to analyze the third replicate of both experiments. For the remaining samples, at least 100 coccoliths were photographed and measured using ImageJ software (Rueden et al., 2017) for a total of 409 coccoliths (Appendix A Table A1).

Statistical analyses (unpaired *t* tests) have been performed for ∅ and RD$_{coccosphere}$ using GraphPad Prism (version 9.05 for MacOS; GraphPad Software, Inc., USA).

### 2.3.2 Protoplast measurements and POC calculation

*Helicosphaera carteri* cellular POC was estimated from protoplast size, following Menden-Deuer and Lessard (2000):

$$\frac{\text{POC[pg]}}{\text{cell}} = a \times V_{cell}^b, \qquad (2)$$

where $V_{cell}^b$ is the protoplast volume, and $a$ and $b$ are constants depending on the considered species (in this

case: $a = 0.216$ and $b = 0.939$; Menden-Deuer and Lessard, 2000). Protoplast volume (in µm) was calculated as $V_{cell} = (\pi/6) d^2 h$, where $d$ and $h$ represent the short and the long-axis cell diameters (in µm) (Sun and Liu, 2003).

The $V_{cell}$ was obtained by measuring cells from culture samples collected at the $T_{final}$ of each experiment. A quantity of 4 mL TS5 of culture samples was fixed with acidic Lugol solution (40 µL TS6), which dissolves the coccoliths while preserving the protoplast for subsequent measurements. Protoplast size (Θ) data were obtained by analyzing at least 50 photos (collected at the inverted microscope) per sample, with ImageJ software (Rueden et al., 2017) using a custom-made macro (https://github.com/mbordiga/Coccoliths, last access: 27 August 2024; Supplement).

With the same macro, data about protoplast aspect ratio (AR$_{protoplast}$) and roundness (RD$_{protoplast}$) were obtained. As for the coccosphere, due to the high correlation between RD and AR, only data about cellular roundness were reported in this work. The averages of the data used for the calculation and the number of individuals analyzed are provided in the Appendix (Appendix A Table A1).

Changes in Θ and RD$_{protoplast}$ have been compared using an unpaired *t* test on GraphPad Prism (version 9.05 for MacOS; GraphPad Software, Inc., USA).

## 3 Results

### 3.1 Coccolith morphology

The analyses at the SEM reveal a non-significant change (*t* test *p* value > 0.05) in the proportion of malformed coccoliths moving from ∼ 295 to 600 µatm of $CO_2$. However, while at the lower $pCO_2$, the species shows almost no malformations (0.66 ± 0.58 %) in the second treatment the malformed coccoliths account for an average of 10.65 ± 10.82 % (Table 2, Fig. 2). The percentage of malformed coccoliths at 600 µatm is characterized by a high standard deviation (SD). On the contrary, at 295 µatm, SD is quite low in all the considered categories (Table 2).

None of the observed samples shows extremely malformed coccoliths. A rough estimation of the number of collapsed coccospheres per sample indicates a percentage far below 1 %. Therefore, a specific count for this category has not been performed, because it is not meaningful. The saturation state of seawater with respect to calcite ($\Omega_{calcite}$) was lower at 600 µatm than at pre-industrial $CO_2$ levels. However, the values were always > 1, indicating that the system was never undersaturated; indeed no dissolution was detected (Table 1).

### 3.2 Coccosphere and protoplast geometry

Cellular POC returns an average of 108.14 ± 5.42 pg per cell at 295 µatm and 118.51 ± 6.41 pg per cell at 600 µatm of $CO_2$, with an unpaired *t* test showing no significant

**Table 2.** Percentages of counted coccoliths at the two different $CO_2$ concentrations. Data reported are averages of three replicates. SD: standard deviation.

| Experiment | $CO_2$ [µatm] | Normal | Malformed | Total no. of counted coccoliths |
|---|---|---|---|---|
| 1 | 295 | 99.34 | 0.66 | 304 |
| SD | | 2.08 | 0.58 | |
| 2 | 600 | 89.35 | 10.65 | 316 |
| SD | | 13.87 | 10.82 | |

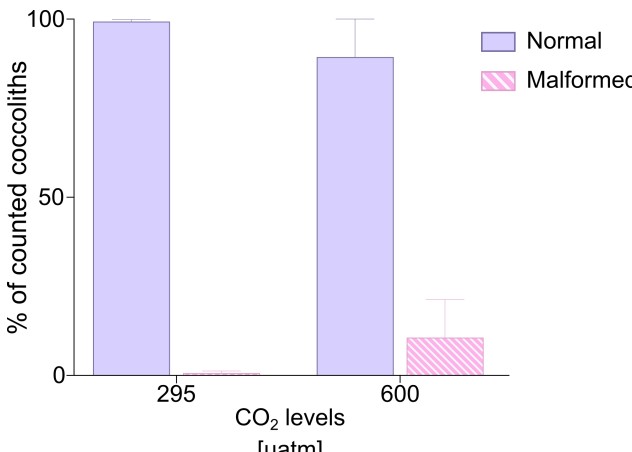

**Figure 2.** Percentages (%) of normal and malformed coccoliths of *H. carteri*. Values reported represent the averages of the three replicates. Error bars show standard deviation.

change between $CO_2$ levels (*t* test *p* value > 0.05; Table 3). A non-significant change is also observed in cellular PIC and in the PIC : POC ratio, showing an average value of $151.86 \pm 4.23$ pg per cell at 295 µatm and $149.47 \pm 9.49$ per
5 cell at 600 µatm of $CO_2$ (*t* test *p* value > 0.05; Table 3) and of $1.37 \pm 0.072$ at 295 µatm and $1.27 \pm 0.013$ at 600 µatm of $CO_2$, respectively (*t* test *p* value > 0.05; Table 3).

The *Helicosphaera carteri* protoplast ($0.90 \pm 0.02$ and $0.90 \pm 0.01$ at 295 and 600 µatm, respectively) and cocco-
10 sphere ($0.89 \pm 0.02$ and $0.88 \pm 0.003$ at 295 and 600 µatm, respectively) roundness do not show any significant variation with increasing $CO_2$ (*t* test *p* value > 0.05), indicating the maintenance of a constant shape at different $CO_2$ levels (Fig. 3a, b; Appendix A Table A1). No changes have
15 been detected for protoplast ($11.45 \pm 0.19$ µm at 295 µatm and $11.81 \pm 0.27$ µm at 600 µatm; *t* test *p* value > 0.05; Fig. 3c; Appendix A Table A1) and coccosphere size ($18.18 \pm 0.25$ µm and $17.92 \pm 0.66$ at 295 and 600 µatm, respectively; *t* test *p* value > 0.05; Fig. 3d; Appendix A Ta-
20 ble A1).

## 4 Discussion

### 4.1 Morphology in *H. carteri* in response to $CO_2$ increase

In the recent years, several studies have focused on coccolithophores' responses under increasing $CO_2$ levels, demon- 25
strating that different species, and often different strains of the same species, exhibit a specific, at times contrasting, response to seawater carbonate chemistry (e.g., Bach et al., 2015; Diner et al., 2015; Langer et al., 2006, 2009, 2011; Langer and Bode, 2011; Müller et al., 2015). These non- 30
uniform results have highlighted the need to analyze the $CO_2$ influence on both coccolithophore species and strains to better predict the whole group reaction to future climate change.

To evaluate the coccolithophore response under high $CO_2$, a key but sometimes neglected parameter is the degree of 35
coccolith malformation and data on morphometrics. To date, few studies have evaluated coccolith morphology (i.e., normal, malformed, or incomplete coccoliths) under seawater carbonate chemistry changes not only in a qualitative way but also in a quantitative way (e.g., Bach et al., 2011, 2012; 40
De Bodt et al., 2010; Diner et al., 2015; Kottmeier et al., 2022; Langer et al., 2006, 2011), but none of them considered the species *H. carteri*. Coccolith morphology is central to ecological and evolutionary success of coccolithophores and is often more telling than calcite production when ques- 45
tions concern the biology, as opposed to the biogeochemistry, of these algae (Henriksen et al., 2003; Langer et al., 2011, 2021; Walker et al., 2018).

In this work, for the first time, we show that the percentage of malformed coccoliths in *H. carteri* does not change 50
in a significant way moving from 295 to 600 µatm of $CO_2$. However, when comparing our findings for *H. carteri* with previous works conducted on other species, it is evident that for most species and strains the percentage of malformed coccoliths at $CO_2$ levels similar to 295 µatm ($\pm 100$ µatm) is 55
higher (Fig. 4a, b). Specifically, a greater percentage of malformed coccoliths (considering all the categories defined by the authors) was observed in six different strains of *E. huxleyi* (RCC1238, RCC1216, RCC1256, RCC1212, B92/11, AC481), four strains of *C. leptoporus* (AC365, RCC1135, 60
RCC1141, RCC1168), and one strain (AC400) of *C. pelagicus* (Fig. 4a, b).

**Table 3.** Data of *H. carteri* cellular PIC and POC obtained from geometry data. Values reported are averages of the replicates. SD: standard deviation.

| CO$_2$ [µatm] | | 295 | 600 | *p* value |
|---|---|---|---|---|
| PIC [pg per cell] | Mean | 151.86 | 149.47 | 0.7755 |
| | SD | 4.23 | 9.49 | |
| POC [pg per cell] | Mean | 108.14 | 118.51 | 0.1000 |
| | SD | 5.42 | 6.41 | |
| PIC : POC | | 1.37 | 1.27 | 0.09595 |
| | SD | 0.072 | 0.013 | |

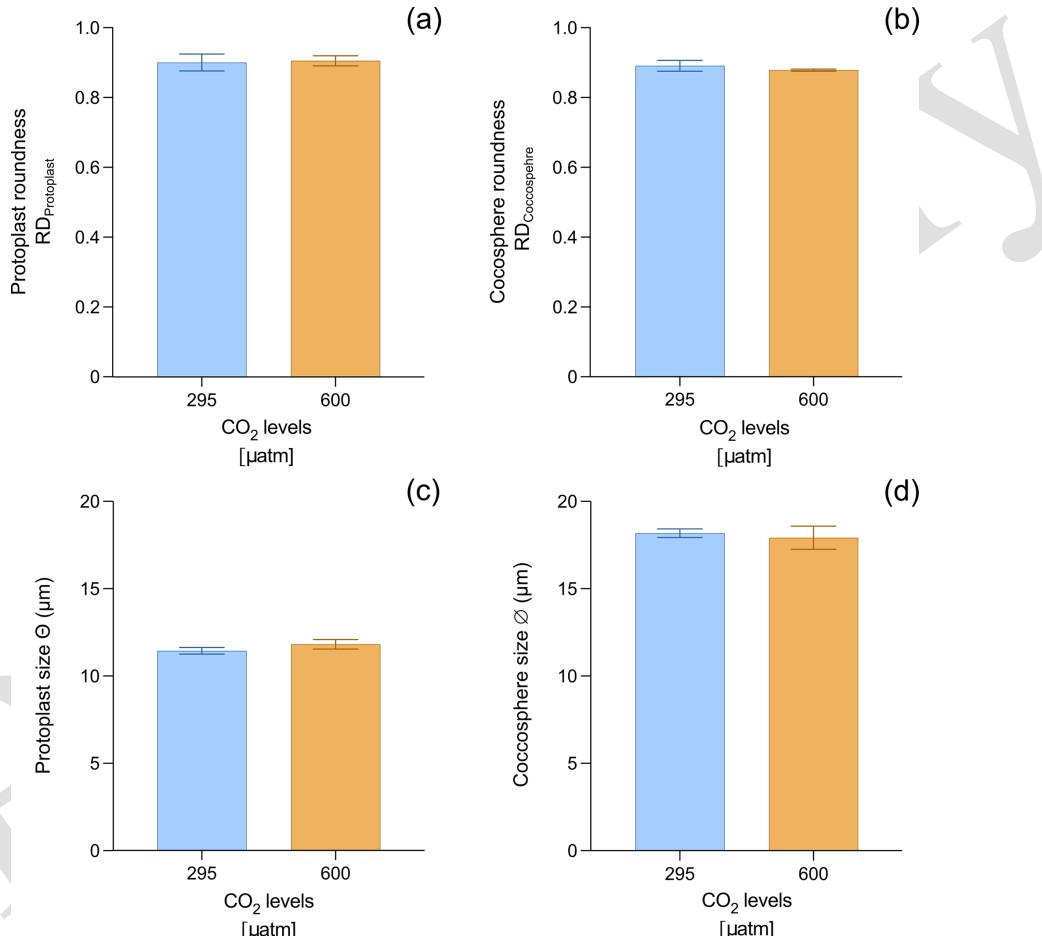

**Figure 3.** Data of *H. carteri* roundness and size, measured on the protoplast **(a, c)** and coccosphere **(b, d)**. Reported values are averages of three replicates. Error bars show standard deviation.

Here we will briefly discuss an issue that distinguishes C-system experiments from other standard culture experiments, namely the fact that the C system is not one single parameter but multiple (see Table 1), as opposed to experiments study-
5 ing the effects of temperature for instance. Different methods for changing the C system are available, i.e., DIC manipulation, TA manipulation, and combined TA-DIC manipulation (Hoppe et al., 2011; Langer and Bode, 2011). Only the lat-

ter method allows for an identification of the parameter of the C system affecting organisms (Langer and Bode, 2011). 10 Very few studies have used this method, and it was found that CO$_2$ and pH are parameters of the C system that affect coccolithophores in typical OA studies (Bach et al., 2011; Langer and Bode, 2011). Here we used DIC manipulation resulting in a so-called coupled C system, as opposed to the 15 decoupled C system obtainable only in combined TA-DIC

manipulation experiments. A coupled C system features correlations between pH, CO$_2$, and CO$_3^{2-}$. It is therefore not possible to distinguish, e.g., pH and CO$_2$ effects. Please note that when we discuss "CO$_2$ effects" we do not literally mean CO$_2$ effects but coupled C-system effects. We have decided to use the shorthand "CO$_2$ effects" because it is common in the literature to do so. Using the strictly correct expression C-system effects has the disadvantage of decreasing readability substantially because a typical phrasing such as "C-system increase/decrease" does not make sense, whereas it does make sense if a single parameter is used as a stand-in for the whole C system.

When considering responses to CO$_2$ levels close to 600 µatm, the percentages of malformed coccoliths in *E. huxleyi* (RCC1238 and RCC1256) are lower than *H. carteri* (Fig. 4a, c). In contrast, *E. huxleyi* (B92/11) and the heavily calcified species *C. leptoporus* (RCC1168) and *C. quadriperforatus* (RCC1141) consistently show a higher percentage of malformed coccoliths compared to *H. carteri* ($\sim$ 60 %–90 %, Fig. 4a, c). Today *C. leptoporus* and *C. quadriperforatus* are mostly considered separate species (https://roscoff-culture-collection.org/rcc-strain-details/1141, Vaulot et al., 2004 TS7), although some authors prefer to consider *C. quadriperforatus* a subspecies (https://www.mikrotax.org/Nannotax3, Young et al., 2022 TS8; Young and Ziveri, 2000 TS9). For a detailed discussion of the taxonomical status of *Calcidiscus*, see Geisen et al. (2004).

The comparison of malformations in different strains/species at one single CO$_2$ level is instructive but not sufficient to assess C-system effects. Malformations in coccolithophores vary both between strains/species and over time in a single strain under constant environmental conditions (Langer et al., 2009, 2013; Langer and Benner, 2009). A better assessment of C-system effects on coccolithophores is achieved when comparing trends of different experiments rather than absolute values of different experiments (Hoppe et al., 2011). Such a comparison clearly suggests species specific responses to CO$_2$, identifying more/less sensitive species (Fig. 4b, c; Diner et al., 2015; Hoppe et al., 2011; Langer et al., 2006, 2011; Langer and Bode, 2011). We are thus confident in saying that the strains RCC1238 and RCC1256 of *E. huxleyi* and RCC1323 of *H. carteri* are less sensitive to acidification than *E. huxleyi* B92/11 and *Calcidiscus*.

However, it is important to note that different authors have observed varying responses among different strains of both *E. huxleyi* and *C. leptoporus*, indicating the absence of a uniform species-specific behavior, potentially linked to genotypic diversity (see Diner et al., 2015; Langer et al., 2009). These diverse responses could be identified in *H. carteri* too. Therefore, additional studies considering different strains of *H. carteri* will be required to identify if our evidence is strain-specific or it can be extended to species level.

## 4.2 *Helicosphaera carteri* sensitivity towards CO$_2$ increase

Studies on coccosphere and protoplast geometry (e.g., $\Theta$, $C_L$, $C_N$) of *H. carteri* strain RCC1323 have been conducted before (Le Guevel et al., 2024; Sheward et al., 2017; Šupraha et al., 2015). However, none of these studies considered the variations in protoplast or coccosphere shapes. In this study, for the first time we show the absence of any significant variation in RD$_{protoplast}$ and RD$_{coccosphere}$ with increasing CO$_2$ (Fig. 3a, b). These results could indicate that the species shape does not depend on CO$_2$ concentrations. Daily observation of the living culture under a light microscope showed in both CO$_2$ treatments that *H. carteri* remained in a good condition, with good motility of the cells. These observations combined with the lack of a CO$_2$ effect on roundness and the small effect on coccolith morphology point to a weak sensitivity of *H. carteri* to seawater acidification/CO$_2$ increase (Figs. 2, 3a, b).

In our study, we also examined the variations in the protoplast and coccosphere geometry ($C_N$, $C_L$, coccosphere and protoplast size) in response to an increase in CO$_2$, observing no significant changes from 295 to 600 µatm (Fig. 3, Appendix A Table A1). Since *H. carteri* $\Theta$, $C_L$, and $C_N$ did not change between our experiments, the cellular POC and PIC content and PIC : POC ratio did not show any substantial variation with increasing CO$_2$ (Fig. 3c, d; Table 3, Appendix A Table A1).

The maintenance of a stable PIC : POC ratio in the same *H. carteri* strain and at similar CO$_2$ levels (300 and 600 µatm) has also recently been observed by Le Guevel et al. (2024) (Fig. 5), who also recorded a slight increase in the coccosphere size within this CO$_2$ range (+0.69 µm from 200 to 600 µatm). These authors grew the species under even higher CO$_2$ levels, recording a decrease in the coccosphere size (−1.05 µm) and moving from 600 to 1400 µatm of CO$_2$. However, this decrease in coccosphere size with increasing CO$_2$/decreasing pH was not associated with a significant trend in the PIC : POC ratio (Le Guevel et al., 2024).

Similar results were documented also in the fossil record by Šupraha and Henderiks (2020), who estimated the PIC : POC ratio for the genus *Helicosphaera* over the last 15 million years (Myr) from the lateral cross-sectional aspect ratio of a coccolith (AR$_L$), following McClelland et al. (2016). These authors documented a stable PIC : POC ratio of this genus along with a reduction of coccolith (and coccosphere) size in response to the global decreasing trend in CO$_2$, which ranged from $\sim$ 350–500 ppm during the Middle Miocene to $\sim$ 200 ppm in the Pleistocene (Herbert et al., 2016; Sosdian et al., 2018; Super et al., 2018; Zachos et al., 2001; Zhang et al., 2013). Šupraha and Henderiks (2020) attributed the lack of change in the ratio between calcification and photosynthesis to the obligate calcifier nature of the genus *Helicosphaera*.

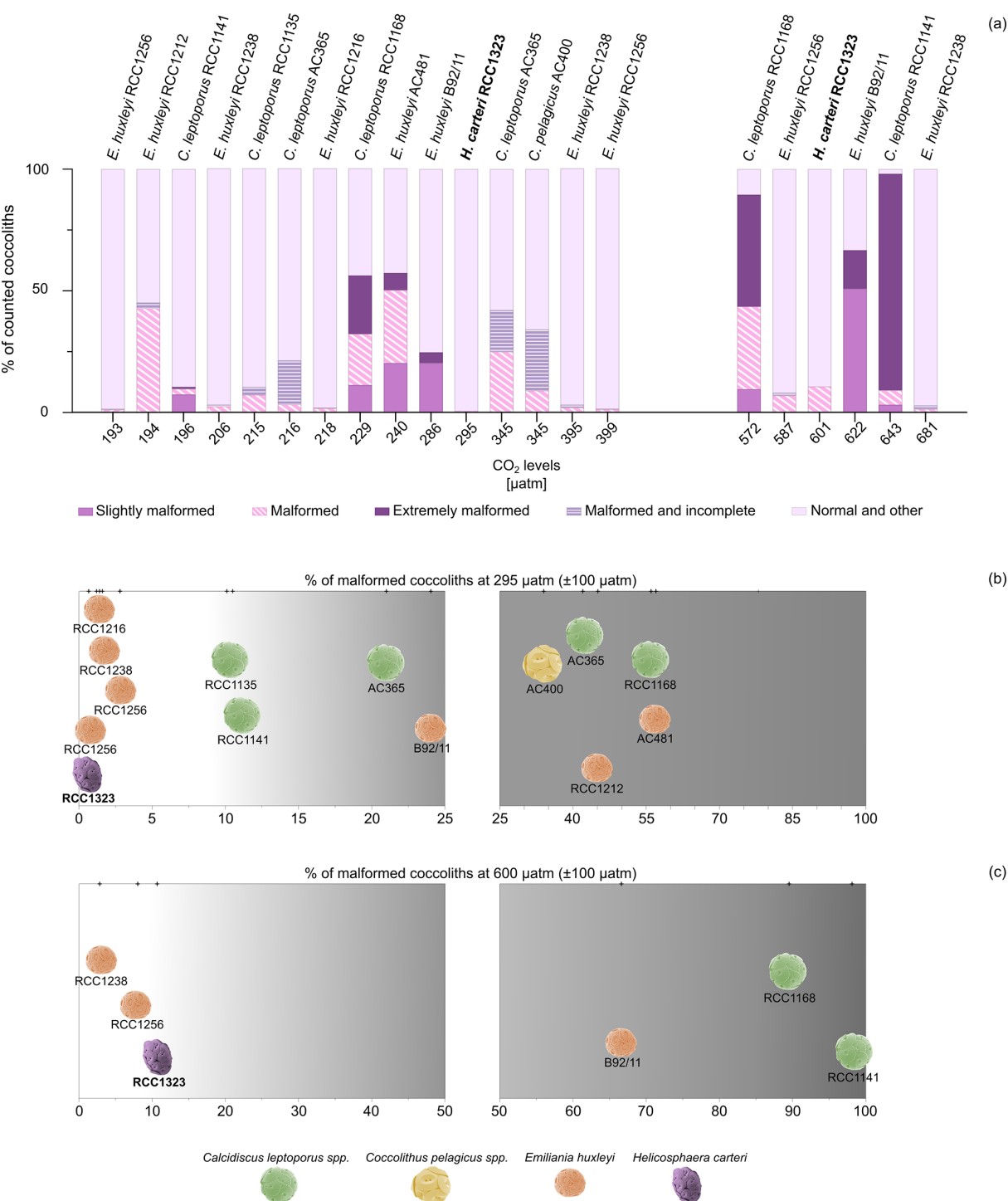

**Figure 4. (a)** Percentages of malformed coccoliths in *H. carteri* (in bold; this work) compared to other species (from the literature) at CO₂ levels close to 295 µatm (±100 µatm) and 600 µatm (±100 µatm). "Other" includes other categories used by the authors, such as fragmented coccoliths or incomplete coccoliths without malformations. **(b–c)** Distribution of the considered strains according to a gradient of increasing percentage of malformation at 295 µatm **(b)** and 600 µatm **(c)**. Different scales have been used. Coccosphere photos are modified from Nannotax.org. Data for comparison include *E. huxleyi* RCC1216, RCC1238, RCC1256, and RCC1212 from Langer et al. (2011); *C. leptoporus* RCC1141 and *C. quadriperforatus* RCC1168 from Diner et al. (2015); *E. huxleyi* AC481 from De Bodt et al. (2010); *C. leptoporus* RCC1135 from Langer and Bode (2011); *C. leptoporus* AC365 and *C. pelagicus* AC400 from Langer et al. (2006); and *E. huxleyi* B92/11 from Bach et al. (2011).

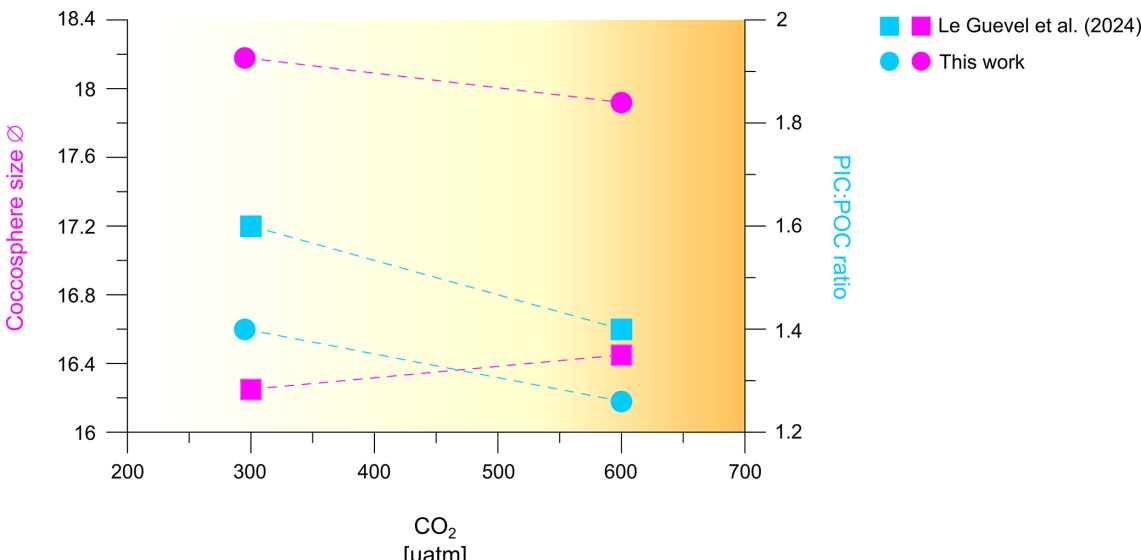

**Figure 5.** Comparison of coccosphere size and PIC : POC ratio of *H. carteri* under increasing CO$_2$, measured in this work and in Le Guevel et al. (2024).

Some coccolithophores such as *Coccolithus braarudii* are obligate calcifiers (i.e., they need to calcify), whereas others such as *Emiliania huxleyi* are facultative calcifiers (i.e., they do not necessarily need to calcify) (Walker et al., 2018). As per our own observation, and the extensive observational record available at the RCC Roscoff (https://roscoff-culture-collection.org/, Vaulot et al., 2004 TS10; Ian Probert, personal communication, 2024) *H. carteri* is an obligate calcifier which might imply a stable PIC : POC ratio because a complete coccosphere is essential for survival (Šupraha and Henderiks, 2020; Walker et al., 2018).

The obligate calcifier-nature of *Helicosphaera* represented by the maintenance of a stable PIC : POC observed both under experimental conditions; (this work, Le Guevel et al., 2024) and in the fossil record (Šupraha and Henderiks, 2020), could represent an advantage in future oceans where the species could stabilize the future C-cycle despite changes in CO$_2$ concentrations. However, to confirm this hypothesis, studies on fossil material deposited during paleo-analogues of future CO$_2$ rise above 600 µatm are required. Reconstructing different coccolithophore species' PIC : POC ratio during past climate events is, indeed, a fundamental tool to better predict their response also to future climate changes. Unfortunately, the chances to find entirely preserved coccospheres in the fossil record is relatively low (Henderiks, 2008). Thus, combining culture studies on PIC : POC estimates from coccosphere, protoplast, AR, and coccolith measurements with observations conducted on fossil coccoliths represents a key tool for investigating the species-specific contribution to the organic C fixation and calcite production in the fossil record, improving our knowledge on the inorganic–organic C balance in the oceans.

With regard to the relationships between PIC : POC ratio and CO$_2$ sensitivity of different species and strains, one of the most significant and consistent evidence is that coccolithophore species with a higher PIC : POC ratio such as *C. leptoporus* (2.08) and *G. oceanica* (1.25) should be more sensitive to increasing CO$_2$ compared to species with lower average PIC : POC ratio such as *E. huxleyi* (0.67), *Syracosphaera pulchra* (0.19), and *Umbilicosphaera sibogae* (0.62; Gafar et al., 2019b). The latter authors hypothesize that a high PIC : POC ratio produces a high cellular proton load that is particularly harmful under ocean acidification conditions. More recently a cellular mechanism underpinning the hypothesis of Gafar et al. (2019b) was proposed (Kottmeier et al., 2022). This cellular mechanism involves Hv-type plasma-membrane proton channels which close under ocean acidification conditions therewith preventing proton export out of the cell with cytosolic acidification ensuing. The low sensitivity of species with lower PIC : POC ratio, like *E. huxleyi*, is confirmed by the comparison in Fig. 4b, c, where *E. huxleyi* appears more resilient in terms of malformations to increasing CO$_2$ levels, compared to both *H. carteri* and *C. leptoporus*. As for *H. carteri* RCC1323, the range of PIC : POC ratio considered by Gafar et al. (2019b), based on data from Šupraha et al. (2015), spans 2.29 to 2.30, and thus there is a relatively high ratio leading to a first inference that this strain may be highly sensitive to CO$_2$ increase. However, recent data about *H. carteri* PIC : POC ratio documented a lower PIC : POC values for this strain (1.27–1.37 ratio this work; ~ 1.4–1.6 ratio Le Guevel et al., 2024) (Fig. 5). The identification of a lower PIC : POC ratio for *H. carteri* RCC1323 (average ratio 1.8) could explain our data documenting a low sensitivity of this strain to increasing CO$_2$, compared to *C.*

*leptoporus* (average ratio 2.08) and *C. quadriperforatus* (average ratio 2.01) (see Sect. 4.1; Diner et al., 2015; Gafar et al., 2019b).

However, in the literature there are sometimes contrasting results on coccolithophore sensitivity towards $CO_2$ in relation to the PIC : POC ratio. Langer et al. (2009), while testing different strains of *E. huxleyi* grown under varying seawater chemistry conditions, documented that the strain with the highest PIC : POC ratio (RCC1216; maximum PIC : POC value 1.2) exhibited the highest percentage of normal coccoliths, corresponding to a low sensitivity towards higher $CO_2$. The most likely explanation for these observations is that the PIC : POC ratio is not a sufficient predictor for the strain's sensitivity to increased $CO_2$. For instance, genetic factors, as suggested by Diner et al. (2015) and Langer et al. (2009), may play a significant role. This once again underscores the importance of analyzing different species and strains and under varying experimental conditions.

## 5 Conclusions

Based on our findings, we can conclude the following:

1. *Helicosphaera carteri*, exposed to pre-industrial $CO_2$ levels and 600 µatm of $CO_2$, shows a low sensitivity to rising $CO_2$, as inferred from protoplast and coccosphere roundness and chiefly from coccolith morphology.

2. The low sensitivity of *H. carteri* to high $CO_2$ is contrasted with the relatively high sensitivity of *Calcidiscus*. An explanation for this surprising species specificity might be the low PIC : POC of *H. carteri* determined here.

3. The PIC : POC ratio of *H. carteri* does not change with changing $CO_2$, suggesting a constant contribution of this species to the rain ratio under ocean acidification.

**Appendix A: Morphometric analyses**

**Table A1.** Summary of *H. carteri* protoplast and coccosphere geometry data obtained from ImageJ software used in this work. The values reported are the averages of the replicates. SD: standard deviation.

| $CO_2$ [µatm] | | 295 | 600 |
|---|---|---|---|
| Coccosphere size $\varnothing$ [µm] | Min | 14.30 | 14.76 |
| | Mean | 18.18 | 17.92 |
| | Max | 21.54 | 23.30 |
| | SD | 0.25 | 0.66 |
| | No. of values | 151 | 158 |
| Protoplast size $\Theta$ [µm] | Min | 9.59 | 9.74 |
| | Mean | 11.45 | 11.81 |
| | Max | 14.12 | 14.56 |
| | SD | 0.19 | 0.27 |
| | No. of values | 161 | 154 |
| Coccosphere aspect ratio $AR_{coccosphere}$ | Min | 1.01 | 1.01 |
| | Mean | 1.13 | 1.14 |
| | Max | 1.46 | 1.3 |
| | SD | 0.02 | 0.005 |
| | No. of values | 151 | 158 |
| Coccosphere roundness $RD_{coccosphere}$ | Min | 0.69 | 0.74 |
| | Mean | 0.89 | 0.88 |
| | Max | 0.99 | 0.99 |
| | SD | 0.02 | 0.003 |
| | No. of values | 151 | 158 |
| Protoplast aspect ratio $AR_{protoplast}$ | Min | 1.00 | 1.01 |
| | Mean | 1.12 | 1.11 |
| | Max | 1.34 | 1.47 |
| | SD | 0.03 | 0.02 |
| | No. of values | 161 | 154 |
| Protoplast roundness $RD_{protoplast}$ | Min | 0.75 | 0.68 |
| | Mean | 0.90 | 0.90 |
| | Max | 0.99 | 0.99 |
| | SD | 0.02 | 0.01 |
| | No. of values | 161 | 154 |
| Coccoliths per coccosphere $C_N$ | Min | 9 | 9 |
| | Mean | 15 | 15 |
| | Max | 20 | 25 |
| | SD | 0.67 | 0.44 |
| | No. of values | 167 | 178 |
| Coccolith length $C_L$ [µm] | Min | 6.20 | 6.6 |
| | Mean | 8.65 | 8.60 |
| | Max | 10.01 | 10.33 |
| | SD | 0.07 | 0.04 |
| | No. of values | 201 | 207 |
| Cellular PIC [pg per cell] | Mean | 151.86 | 149.47 |
| | SD | 4.23 | 9.50 |
| | No. of values | 201 | 207 |
| Cellular POC [pg per cell] | Mean | 108.14 | 118.51 |
| | SD | 5.42 | 6.41 |
| | No. of values | 161 | 154 |

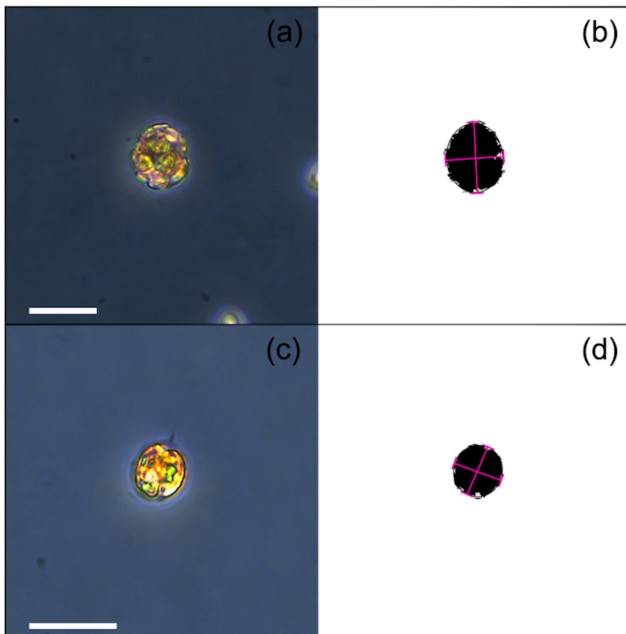

**Figure A1.** Photo of a coccosphere **(a, b)** and a protoplast **(c, d)** before and after ImageJ processing. Measurements of the long and short axes are indicated in pink **(c, d)**. All bars are 20 µm.

*Code and data availability.* The Java script used here within ImageJ software is available on GitHub: https://github.com/mbordiga/Coccoliths (last access: TS11; https://doi.org/10.5281/zenodo.15122548, mbordiga, 2025).

*Supplement.* The supplement related to this article is available online at [the link will be implemented upon publication].

*Author contributions.* SB, GL, and MB conceived the study; FC and MB designed the experiments, and MB carried out the experiments; SB and GL analyzed the coccolithophore samples, the data sets, and elaborated the data; SB wrote the first draft with contributions on data discussion and interpretation from GL, MB, and CL; MB, CL, and ADG provided financial support for the project; and PZ and ADG provided a critical review. All authors contributed to the final draft.

*Competing interests.* The contact author has declared that none of the authors has any competing interests.

*Acknowledgements.* We thank Maria Pia Riccardi and Maya Musa (CISRiC-Arvedi Laboratory, University of Pavia) for technical assistance during SEM analyses and Marina Cabrini, Alfred Beran, Federica Relitti, and Vincenzo Alessandro Laudicella (OGS, Trieste) for their technical support.

*Financial support.* This work was funded by MUR for ECORD-IODP Italia 2018 to Manuela Bordiga within the project "Geochemistry and marine biology united to refine climate models", conducted at the National Institute of Oceanography and Applied Geophysics (OGS), and for the Italian national inter-university PhD course in Sustainable Development and Climate change (https://www.phd-sdc.it, last access: 10 January 2025) to Stefania Bianco. The project was also supported by Claudia Lupi and Andrea Di Giulio with University of Pavia Research Funds (FAR 2021–2023) and by the Okada-McIntyre Graduate Research Fellowship of INA awarded to Stefania Bianco. Gerald Langer was funded by the Spanish Ministry of Universities TS12 through a Maria Zambrano grant and the Generalitat de Catalunya (MERS, 2021 SGR00640). This work contributes to ICTA-UAB "María de Maeztu" Programme for Units of Excellence of the Spanish Ministry of Science and Innovation (CEX2019-000940-M) and to the BIOCAL Project (PID2020-113526RB-I00, Spanish Ministry of Science and Innovation). This paper and related research have been conducted during and with the support of the Italian national inter-university PhD course CE2 in Sustainable Development and Climate change (https://www.phd-sdc.it, last access: TS13), and it is part of Stefania Bianco's PhD. TS14

*Review statement.* This paper was edited by Emilio Marañón and reviewed by Austin Grubb and Andres Rigual-Hernandez.

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

**Remarks from the language copy-editor**

CE1    Please confirm whether this is what you mean. Please note that different coloured fonts cannot be displayed in the table.

CE2    This course has already been mentioned above; can the sentence be deleted here?

**Remarks from the typesetter**

TS1    Please check all affiliation codes and affiliations carefully and confirm if they are correct.

TS2    Please confirm citation.

TS3    Please confirm citation.

TS4    Please give an explanation of why this needs to be changed. We have to ask the handling editor for approval. Thanks.

TS5    Please give an explanation of why this needs to be changed. We have to ask the handling editor for approval. Thanks.

TS6    Please give an explanation of why this needs to be changed. We have to ask the handling editor for approval. Thanks.

TS7    Please confirm citation.

TS8    Please confirm changed URL and citation.

TS9    Please confirm citation.

TS10    Please confirm citation.

TS11    Please provide date of last access.

TS12    Please confirm this is "Ministerio de Universidades".

TS13    Please provide date of last access.

TS14    Please confirm both Acknowledgements and Financial support sections. Please also confirm the updated funding information in the MS records.

TS15    Please provide the title.

TS16    Please confirm reference list entry.