# Peer review of "Low sensitivity of a heavily-calcified coccolithophore under increasing CO2: the case study of *Helicosphaera carteri"

_EGUsphere, 2024_

## Referee Comment (RC1)

**Overall**
The aim of the study is worthwhile, as it focuses on a relatively understudied coccolithophore (compared to model species such as *Emiliania/Gephyrocapsa huxleyi* and *Gephyrocapsa oceanica*). The authors investigated whether elevated $pCO_2$ impacts *Helicosphaera carteri*, assessing coccolith morphology and particulate inorganic and organic carbon (PIC and PIC, respectively). The authors claim that the results of this study suggest that *H. carteri* may have a constant contribution to the rain ratio under ocean acidification.

However, there are major weaknesses in how the data is presented and interpreted (or not used in the discussion) that lead me to recommend that this version of the manuscript be rejected.

First, the authors only include the impact of $pCO_2$ levels in the interpretation of the data, excluding the rest of the carbonate chemistry data presented in Table 1. Since coccolithophores are particularly dependent on carbonate chemistry, this oversight significantly detracts from the rest of the manuscript. See comments in Discussion for more details.
Second, the authors do not accurately represent the results of statistical analyses on a number of occasions in the Results. There are also occasions where a sentence contradicts a previous statement. This needs to be corrected. See comments in Results section for specifics.
Lastly, the authors use figures/tables that present the same data repeatedly. This does not add evidence to support their interpretation of the data. It would be better for the authors to choose which figure/table best presents the data and eliminate the other.

**Abstract**
Line 23-25: The authors state "In this study...whether high $pCO_2$/low pH does affect the morphology of *H. carteri* coccoliths...". But again, a central weakness of the manuscript is that the results and discussion only focus on the $pCO_2$, ignoring the rest of the carbonate chemistry.

**Introduction**
There are some word choice and grammatical issues, but overall, the introduction does a good job providing the rationale for the experiments.

**Materials and Methods**
*Carbonate chemistry*:
- DIC levels decreased under elevated $CO_2$. To better replicate OA conditions, wouldn't it be better for DIC levels to remain similar (or even increase) under elevated $CO_2$ compared to the control $CO_2$ condition?
- Table 1: This should be in the results. The atmospheric $CO_2$ levels influence the carbonate chemistry, which can impact coccolith morphology. In addition, the

standard deviation for pH is given. What is the error for the variables (DIC, TA, etc...)? This results here are underutilized throughout the rest of the manuscript.

Minor:

- Be sure to include the manufacturer info for each instrument.

Why not directly measure PIC and POC?

**Results**

- Lines 199-205: The authors state that there was a "slight change in the proportion of malformed coccoliths ~295 and 600 μatm of $CO_2$" (Line 199). This is not supported by the data presented. The average ± standard deviation percentage of malformed coccoliths are not different between the 295 and 600 μatm $CO_2$ treatments [other statistics (e.g., unpaired t-tests) are not provided]. The authors can still highlight the high variability of malformed coccoliths in the 600 μatm $CO_2$ treatment.
- Figure 2: The data presented here are misleading with the standard deviation not depicted, which would show overlap between the two $CO_2$ treatments. Additionally, these data are already presented in Table 2. It is unnecessary to show the same data in a Table and a Figure. The authors should decide which presentation of the data is best for the manuscript.
- Line 205-207: The authors state that "All coccosphere were intact" but then use the next two sentences to mention the small number of collapsed coccospheres detected. This is contradictory. These lines should be edited to resolve this contradiction.
- Lines 220-225: The authors state there are changes in cellular POC, cellular PIC, and PIC/POC, but then state that the changes are not statistically significant. This is contradictory. Just state that there was no significant difference between the two $CO_2$ treatments. Also, remove the methods for the unpaired t-tests. It is already in the Materials and Methods.
- Line 227-8: What are the units? Include the units for the values.
- Line 229-232: Again, the authors use "a non-significant change" instead of stating that 'no change/difference was detected' between protoplast and coccolith size.

I recommend adjusting how data are presented in the text by changing specific references to the following format: 'average±SD (t-test p-value<X; Table X)'.

**Discussion**

*Major comments*:

The authors only include the impact of atmospheric $CO_2$ levels in the interpretation of the data, excluding the rest of the carbonate chemistry data presented in Table 1. Coccolithophores require $HCO_3^-$ as a substrate for calcification. The authors show a drop in pH and $[HCO_3^-]$ when $CO_2$ increased from 295 to 600 μatm (Table 1), but do not include these variables when interpreting the data. This leads to an incomplete interpretation of the data. What about the lower pH? Is it possible that the variability in malformed

coccoliths at 600µatm $CO_2$ is due to the combination of lower pH (unfavorable for calcite precipitation), lower $HCO_3^-$ concentration (substrate for calcification), and Omega > 1 (calcite formation still slightly favored)?

*Some line-by-line comments*:
Line 254-255: This statement is not true. See comments on Results for details.

Line 266-274: I don't understand the point of Lines 266-270.

Line 272: "...the negative effect of carbonate chemistry". This phrase does not make sense.

Line 275-276: "the effect of proton inhibition" The authors do not present any data on protons (e.g., pH) from the previous work cited.

Line 278-279: "less sensitive to acidification" The authors did not include any carbonate chemistry data (aside from atmospheric $CO_2$) when referencing coccolith malformation documented throughout the literature. It is inappropriate to make a claim about sensitivity to acidification without showing the relevant acidification data.

Figure 4: What about the pH in the other experiments? And other carbonate chemistry parameters (i.e., $HCO_3^-$, $CO_3^{2-}$, DIC, etc...)? It is difficult to interpret comparisons when only $CO_2$ µatm is included since the carbonate chemistry of the growth medium can varying based on the buffering capacity in seawater.
Figure 4b-c: Is this just showing the data from panel a) again?

Figure 5: What were the other relevant conditions in the other study, aside from $pCO_2$?

The rest of the Discussion focuses on comparing the findings in the manuscript to previous work. This section will need to be revised accordingly after the issues identified throughout the manuscript are resolved.

---

## Author Comment (AC1)

[Figure]

(a)

% of counted coccoliths

*E. huxleyi* RCC1256
*E. huxleyi* RCC1212
*C. leptoporus* RCC1141
*E. huxleyi* AC481
*E. huxleyi* RCC1238
*C. leptoporus* RCC1135
*C. leptoporus* AC365
*E. huxleyi* RCC1216
*C. leptoporus* RCC1168
*E. huxleyi* B92/11
**H. carteri RCC1323**
*C. leptoporus* AC365
*C. pelagicus* AC400
*E. huxleyi* AC481
*E. huxleyi* RCC1238
*E. huxleyi* RCC1256

*C. leptoporus* RCC1168
*E. huxleyi* RCC1256
**H. carteri RCC1323**
*E. huxleyi* B92/11
*C. leptoporus* RCC1141
*E. huxleyi* RCC1238

CO₂ levels [μatm]

193, 194, 196, 201, 206, 215, 216, 218, 229, 286, 295, 345, 345, 389, 395, 399, 572, 587, 601, 621, 643, 681

Legend: Slightly malformed | Malformed | Extremely malformed | Malformed and incomplete | Normal and other

(b) % of malformed coccoliths at 295 μatm (±100 μatm)

RCC1216, RCC1238, RCC1135, AC365, RCC1256, RCC1141, RCC1256, RCC1323, B92/11
AC365, AC400, RCC1168, AC481, AC481, RCC1212

(c) % of malformed coccoliths at 600 μatm (±100 μatm)

RCC1238, RCC1256, RCC1323
RCC1168, B92/11, RCC1141

*Calcidiscus leptoporus spp.* | *Coccolithus pelagicus spp.* | *Emiliania huxleyi* | *Helicosphaera carteri*

---

## Author Response (AR1)

**REPLY to Assigned Editor:**

We have addressed all points raised by the reviewers and the editor and have revised our manuscript accordingly. We disagree with the reviewers in some minor points and have explained why we do so in our rebuttal. We are grateful for the reviewer's insightful comments which have helped us to improve the manuscript. The reviewers and the associate editor highlight missing statistics and insufficient explanation regarding both the experimental setup (in particular DIC concentrations) and the interpretation of data (in particular the potential influence of different parameters of the C-system). We have added the statistics and explained our setup and interpretation in more detail.

DIC levels decreased under elevated $CO_2$. To better replicate OA conditions, wouldn't it be better for DIC levels to remain similar (or even increase) under elevated $CO_2$ compared to the control $CO_2$ condition?
**REPLY: We agree with the reviewer in so far that OA conditions do not feature decreased DIC concentrations. However, DIC is not the parameter of the C-system affecting coccolithophores in typical OA studies (Bach et al., 2011; Langer and Bode, 2011). Under DIC concentrations below ca 1000uM, DIC and/or bicarbonate ion concentration might play a role too (Buitenhuis, 1999). In our experiment the lowest DIC is ca 1400 µM and the highest ca 1700 µM. Within this range in DIC, the difference of ca 300 µM between treatments does not produce measurable effects. The parameters of the C-system that will have affected *H. carteri* most likely are either pH or $CO_2$ (Bach et al., 2011; Langer and Bode, 2011); a possible but unlikely candidate is carbonate ion concentration. All three parameters fall within the range of typical OA studies (e.g., Bach et al., 2011; Hoppe et al., 2011; Langer et al., 2009; Langer and Bode, 2011; Milner et al., 2016; Zondervan et al., 2002). Therefore, our experimental setup is suitable for our purpose.**

**We added the following to the Material and Methods section:**

"Typical OA scenarios do not feature decreasing DIC concentrations. In our experiment the lowest DIC is ca 1400 µM (high $CO_2$, low pH) and the highest ca 1700 µM (low $CO_2$, high pH, Table 1). Despite this atypical $CO_2$-DIC combination for OA scenarios the latter does not undermine the suitability of our experimental setup because DIC is not the parameter of the C-system affecting coccolithophores in typical OA studies (Bach et al., 2011; Hoppe et al., 2011; Langer and Bode, 2011). Only under DIC concentrations below ca 1000 µM, DIC and/or bicarbonate ion concentration might play a role too (Buitenhuis, 1999). The parameters of the C-system that will have affected *H. carteri* most likely are either pH or $CO_2$ (Bach et al., 2011; Langer and Bode, 2011); a possible but unlikely candidate is carbonate ion concentration. All three parameters fall within the range of typical OA studies (e.g., Bach et al., 2011; Hoppe et al., 2011; Johnson et al., 2022; Kottmeier et al., 2022; Langer et al., 2009; Langer and Bode, 2011; Milner et al., 2016; Zondervan et al., 2002). Therefore, our experimental setup is suitable for our purpose."

**We also corrected an error made in Discussion section 4.2 at line 430: OLD TEXT:** A non-significant variation in coccosphere size and PIC:POC ratio in the same *H. carteri* strain and at similar $CO_2$ levels (300 µatm and 600 µatm) has recently been observed also by Le Guevel et al. (2024) (Fig. 5).

**NEW TEXT:** A non-significant variation in PIC:POC ratio in the same *H. carteri* strain and at similar $CO_2$ levels (300 µatm and 600 µatm) has recently been observed also by Le Guevel et al. (2024) (Fig. 5).

**References**

Bach, L. T., Riebesell, U., and Schulz, K. G.: Distinguishing between the effects of ocean acidification and ocean carbonation in the coccolithophore Emiliania huxleyi, Limnology & Oceanography, 56, 2040–2050, 400 https://doi.org/10.4319/lo.2011.56.6.2040, 2011.

Buitenhuis, E. T., De Baar, H. J. W., and Veldhuis, M. J. W.: Photosynthesis and calcification by Emiliania huxleyi (Prymnesiophyceae) as a function of inorganic carbon species, J. Phycol., 35, 949–959, https://doi.org/10.1046/j.1529-8817.1999.3550949.x, 1999.

Hoppe, C. J. M., Langer, G., and Rost, B.: Emiliania huxleyi shows identical responses to elevated $pCO_2$ in TA and DIC manipulations, Journal of Experimental Marine Biology and Ecology, 406, 54–62, https://doi.org/10.1016/j.jembe.2011.06.008, 2011.

Johnson, R., Langer, G., Rossi, S., Probert, I., Mammone, M., and Ziveri, P.: Nutritional response of a coccolithophore to changing PH and temperature, Limnology & Oceanography, 67, 2309–2324, https://doi.org/10.1002/lno.12204, 2022.

Kottmeier, D. M., Chrachri, A., Langer, G., Helliwell, K. E., Wheeler, G. L., and Brownlee, C.: Reduced $H^+$ channel activity disrupts pH homeostasis and calcification in coccolithophores at low ocean pH, Proc. Natl. Acad. Sci. U.S.A., 119, e2118009119, https://doi.org/10.1073/pnas.2118009119, 2022.

Langer, G. and Bode, M.: $CO_2$ mediation of adverse effects of seawater acidification in Calcidiscus leptoporus, Geochem Geophys Geosyst, 12, 2010GC003393, https://doi.org/10.1029/2010GC003393, 2011.

Langer, G., Nehrke, G., Probert, I., Ly, J., and Ziveri, P.: Strain-specific responses of Emiliania huxleyi to changing seawater carbonate chemistry, Biogeosciences, 6, 2637–2646, https://doi.org/10.5194/bg-6-2637-2009, 2009.

Milner, S., Langer, G., Grelaud, M., and Ziveri, P.: Ocean warming modulates the effects of acidification on Emiliania huxleyi calcification and sinking, Limnol. Oceanogr., 61, 1322–1336, https://doi.org/10.1002/lno.10292, 2016.

Zondervan, I., Rost, B., and Riebesell, U.: Effect of $CO_2$ concentration on the PIC/POC ratio in the coccolithophore Emiliania huxleyi grown under light-limiting conditions and different daylengths, J. Exp. Mar. Biol. Ecol., 272, 55–70, https://doi.org/10.1016/S0022-0981(02)00037-0, 2002.

**Rebuttal letter Reviewer 1**

**Overall**

The aim of the study is worthwhile, as it focuses on a relatively understudied coccolithophore (compared to model species such as Emiliania/Gephyrocapsa huxleyi and Gephyrocapsa oceanica). The authors investigated whether elevated $pCO_2$ impacts Helicosphaera carteri, assessing coccolith morphology and particulate inorganic and organic carbon (PIC and PIC, respectively). The authors claim that the results of this study suggest that H. carteri may have a constant contribution to the rain ratio under ocean acidification.

However, there are major weaknesses in how the data is presented and interpreted (or not used in the discussion) that lead me to recommend that this version of the manuscript be rejected.

**REPLY:** We appreciate the reviewer's generally positive assessment of our study and acknowledge that there are shortcomings in interpretation and presentation of the data. The reviewer identifies three main points which we address briefly here (for detailed replies see below).

First, the authors only include the impact of $pCO_2$ levels in the interpretation of the data, excluding the rest of the carbonate chemistry data presented in Table 1. Since coccolithophores are particularly dependent on carbonate chemistry, this oversight significantly detracts from the rest of the manuscript. See comments in Discussion for more details.

**REPLY:** We did not mean to suggest that we identify $CO_2$ as the parameter of the C-system affecting *H. carteri*. We merely used $CO_2$ as a stand-in for the C-system, because we use a so-called coupled C-system. We are grateful for pointing out our misleading phrasing. We clarified this point and have briefly discussed which parameter of the C-system might be the dominant influence on coccolithophores.

Second, the authors do not accurately represent the results of statistical analyses on a number of occasions in the Results. There are also occasions where a sentence contradicts a previous statement. This needs to be corrected. See comments in Results section for specifics.

**REPLY:** We fixed the statistical issues and have resolved contradicting statements.

Lastly, the authors use figures/tables that present the same data repeatedly. This does not add evidence to support their interpretation of the data. It would be better for the authors to choose which figure/table best presents the data and eliminate the other.

**REPLY:** We present data in a slightly redundant way (same data in figure and table) in order to increase readability. We believe that the wide readership of Biogeosciences will appreciate this choice since it considerably facilitates access to central information for the non-specialist.

**Abstract**

Line 23-25: The authors state "In this study...whether high $pCO_2$/low pH does affect the morphology of H. carteri coccoliths...". But again, a central weakness of the manuscript is that the results and discussion only focus on the $pCO_2$, ignoring the rest of the carbonate chemistry.

**REPLY:** In the abstract we use $pCO_2$ as a stand-in parameter for correlated parameters (pH, $CO_3^{2-}$) for better readability. We agree with the reviewer in so far that other parameters of the C-system could affect coccolithophores. This question is an interesting one but not within the scope of our study. We, however, agree that a brief discussion of this question does improve the manuscript and have therefore added the following to the Discussion section (see also reply to comments below):

**NEW TEXT (INSERT AFTER LINE 260:** "Here we will briefly discuss an issue that distinguishes C-system experiments from other standard culture experiments, namely the fact that the C-system is not one single parameter but multiple (see Table 1), as opposed to experiments studying the effects of temperature for instance. Different methods for changing the C-system are available, i.e. DIC manipulation, TA manipulation, and combined TA-DIC manipulation (Hoppe et al., 2011; Langer and Bode, 2011). Only the latter method allows for an identification of the parameter of the C-system affecting organisms (Langer and Bode, 2011). Very few studies have used this method, and it was found that $CO_2$ and pH are parameters of the C-system that affect coccolithophores in typical OA studies (Bach et al., 2011; Langer and Bode, 2011). Here we used DIC manipulation resulting in a so-called coupled C-system, as opposed to the decoupled C-system obtainable only in combined TA-DIC manipulation experiments. A coupled C-system features correlations between pH, $CO_2$, and $CO_3^{2-}$. It is therefore not possible to distinguish e.g. pH and $CO_2$ effects. Please note that when we discuss "$CO_2$ effects" we do not literally mean $CO_2$ effects but coupled C-system effects. We have decided to use the shorthand "$CO_2$ effects" because it is common in the literature to do so. Using the strictly correct expression C-system effects has the disadvantage of decreasing readability substantially because a typical phrasing such as "C-system increase/decrease" does not make sense, whereas it does make sense if a single parameter is used as a stand-in for the whole C-system."

**Introduction**

There are some word choice and grammatical issues, but overall, the introduction does a good job providing the rationale for the experiments.

**REPLY:** We appreciate the reviewer's positive assessment of the Introduction. We fixed the grammatical issues.

**Materials and Methods**

Carbonate chemistry:

- DIC levels decreased under elevated $CO_2$. To better replicate OA conditions, wouldn't it be better for DIC levels to remain similar (or even increase) under elevated $CO_2$ compared to the control $CO_2$ condition?

**REPLY:** We agree with the reviewer in so far that OA conditions do not feature decreased DIC concentrations. However, DIC is not the parameter of the C-system affecting coccolithophores in typical OA studies (Bach et al., 2011; Langer and Bode, 2011). Under DIC concentrations below ca 1000uM, DIC and/or bicarbonate ion concentration might play a role too (Buitenhuis 1999). In our experiment the lowest DIC is ca 1400 uM and the highest ca 1700 uM. Within this range in DIC, the difference of ca 300uM between treatments does not produce measurable effects. The parameters of the C-system that will have affected H. carteri most likely are either pH or $CO_2$ (Bach et al., 2011; Langer and Bode, 2011); a possible but unlikely candidate is carbonate ion concentration. All three parameters fall within the range of typical OA studies (e.g. Bach et al., 2011; Hoppe et al., 2011; Langer et al., 2009; Langer and Bode, 2011; Milner et al., 2016; Zondervan et al., 2002). Therefore, our experimental setup is suitable for our purpose.

- Table 1: This should be in the results. The atmospheric $CO_2$ levels influence the carbonate chemistry, which can impact coccolith morphology. In addition, the standard deviation for pH is given. What is the error for the variables (DIC, TA, etc...)? This results here are underutilized throughout the rest of the manuscript. **REPLY:** We now give the standard deviation for all C-system parameters in Table 1. As stated above we now also discuss the potential impact of C-system parameters other than $CO_2$ in the Discussion section:

**NEW TEXT:** see reply to comment on Line 23-25

| Parameter | Exp. 295 | Exp. 600 |
|---|---|---|
| $CO_2$ (µatm) | 294.56 | 601.5 |
| SD | 17.84 | 59.74 |
| $CO_2$ (µmol/kg) | 9.78 | 19.94 |
| SD | 0.59 | 1.98 |
| $HCO_3^-$ (µmol/kg) | 1413.49 | 1213.70 |
| SD | 106.02 | 144.50 |
| $CO_3^{2-}$ (µmol/kg) | 141.44 | 51.72 |

| | | |
|---|---|---|
| SD | 16.62 | 13.38 |
| **DIC (μmol/kg)** | **1677.50** | **1374.72** |
| **SD** | **140.87** | **142.03** |
| **TA (mmol/kg$^{-1}$)** | **1853.82** | **1452.54** |
| **SD** | **166.93** | **146.41** |
| **pH NBS** | **8.18** | **7.81** |
| **SD** | **0.025** | **0.064** |
| Ω calcite | 3.38 | 1.24 |
| SD | 0.40 | 0.32 |

**Table 1. Parameters of the carbonate system. In black are the values obtained from the CO2SYS program; in bold are the average values directly measured in duplicates per each replica of both the experiments. The average pH values are derived from the whole data collected in continuum along the experiments (pH standard deviation 0.01). SD= standard deviation.**

These results here are underutilized throughout the rest of the manuscript.

Minor:

• Be sure to include the manufacturer info for each instrument.
**REPLY:** We now include the manufacturer info.

Why not directly measure PIC and POC?
**REPLY:** Both geometrical and chemical analyses of PIC and POC are established methods (Langer et al., 2009, Milner et al., 2016; Rosas-Navarro et al., 2018). A direct comparison of these two methods shows that both are equally applicable in coccolithophore studies (Rosas-Navarro et al., 2018).

**Results**

- Lines 199-205: The authors state that there was a "slight change in the proportion of malformed coccoliths ~295 and 600 μatm of $CO_2$" (Line 199). This is not supported by the data presented. The average ± standard deviation percentage

of malformed coccoliths are not different between the 295 and 600 µatm $CO_2$ treatments [other statistics (e.g., unpaired t-tests) are not provided]. The authors can still highlight the high variability of malformed coccoliths in the 600 µatm $CO_2$ treatment.

**REPLY:** We agree with the conclusion that there is no statistically significant change in the percentage of malformed coccoliths. We now include the unpaired t-test (p value=0.1815).

**The text has been adjusted as follows, also following the indications of Reviewer 2.**

**OLD TEXT:** "The analyses at the SEM revealed a slight change in the proportion of malformed coccoliths moving from ~295 to 600 µatm of $CO_2$. Indeed, while at the lower $pCO_2$, the species shows almost no malformations (0.66%), an increase in the percentage of malformed coccoliths is observed in the second treatment, where the normal coccoliths account for an average of 89.35% (Table 2, Fig. 2). The percentage of malformed coccoliths at 600 µatm is characterized by a high standard deviation (SD), suggesting a relatively high variability among the triplicates. On the contrary, at 295 µatm, SD is quite low in all the considered categories, reflecting a greater degree of consistency between the samples compared to 600 µatm (Table 2)."

**NEW TEXT:** "The analyses at the SEM revealed a non-significant change (t-test p value >0.05) in the proportion of malformed coccoliths moving from ~295 to 600 µatm of $CO_2$. However, while at the lower $pCO_2$, the species shows almost no malformations (0.66±0.58%) in the second treatment the malformed coccoliths account for an average of 10.65±10.82% (Table 2, Fig. 2). The percentage of malformed coccoliths at 600 µatm is characterized by a high standard deviation (SD), suggesting a relatively high variability among the triplicates. On the contrary, at 295 µatm, SD is quite low in all the considered categories, reflecting a greater degree of consistency between the samples compared to 600 µatm (Table 2)."

- Figure 2: The data presented here are misleading with the standard deviation not depicted, which would show overlap between the two $CO_2$ treatments. Additionally, these data are already presented in Table 2. It is unnecessary to show the same data in a Table and a Figure. The authors should decide which presentation of the data is best for the manuscript.
  **REPLY:** We agree with the reviewer in so far that stacked bar plots do not allow for error bars and are therefore less informative than scatter plots or tables. Since we also agree that standard deviations are needed, we included them in the Table. For reasons of readability, however, we decided to provide a figure that allows for quick access to data. We realize that this is technically redundant, but we feel that the readers of Biogeosciences, being a diverse readership, will appreciate this choice.

- Line 205-207: The authors state that "All coccosphere were intact" but then use the next two sentences to mention the small number of collapsed coccospheres detected. This is contradictory. These lines should be edited to resolve this contradiction.
  **REPLY:** We amended the text to resolve this contradiction:

  **NEW TEXT:**
  "None of the observed samples showed extremely malformed coccoliths. A rough estimation of the number of collapsed coccospheres per sample indicated a percentage far below 1%. Therefore, a specific count for this category was not performed, because it is not meaningful."

- Lines 220-225: The authors state there are changes in cellular POC, cellular PIC, and PIC/POC, but then state that the changes are not statistically significant. This is contradictory. Just state that there was no significant difference between the two $CO_2$ treatments. Also, remove the methods for the unpaired t-tests. It is already in the Materials and Methods.
  **REPLY:** We now state that there is no significant difference and we removed the methods for the unpaired t-tests.

  **The text has been adjusted as follows:**
  "Cellular POC returns an average of 108.14±5.42 pg cell$^{-1}$ at 295 µatm and 118.51±6.41 pg cell$^{-1}$ at 600 µatm of $CO_2$. The unpaired t-test indicates that moving from the lowest to the highest $CO_2$ level, the cellular POC does not change significantly (t-test p value>0.05; Table 3). A non-significant change is also observed in cellular PIC and in the PIC:POC ratio, showing an average value of 150.66±1.59 pg cell$^{-1}$ (t-test p value>0.05; Table 3) and of 1.32±0.07, respectively (t-test p value >0.05; Table 3)."

  **Table 3 has been adjusted as follows:**

| $CO_2$ [µatm] | | 295 | 600 | p value |
|---|---|---|---|---|
| PIC [pg cell$^{-1}$] | Mean | 151.86 | 149.47 | 0.7755 |
| | SD | 4.23 | 9.49 | |
| POC [pg cell$^{-1}$] | Mean | 108.14 | 118.51 | 0.1000 |
| | SD | 5.42 | 6.41 | |

| PIC:POC | 1.37 | 1.27 | 0.09595 |
|---|---|---|---|
| SD | 0.072 | 0.013 | |

**Table 3. Data of *H. carteri* cellular PIC and POC and PIC:POC obtained from geometry data. Values reported are averages of the replicates. SD = standard deviation.**

- Line 227-8: What are the units? Include the units for the values.
  **REPLY:**
  The values are derived from the ratio between the major and minor axes of the protoplast/coccosphere (µm/µm). Since the unit of both factors is the same, we did not think it was necessary to specify it. However, we have now updated the text and included the unit of measurement.

  **NEW TEXT:**
  **Lines 226-228:** "*Helicosphaera carteri* protoplast (0.90±0.06 µm/µm) and coccosphere (0.89±0.05 µm/µm) roundness does not show any significant variation with increasing $CO_2$ (t-test p value>0.05), indicating the maintenance of a constant shape at different $CO_2$ levels (Fig. 3 a, b; Appendix A Table A1)."

- Line 229-232: Again, the authors use "a non-significant change" instead of stating that 'no change/difference was detected' between protoplast and coccolith size.
  **REPLY: The text has been adjusted as follows:**
  "No changes have been detected for protoplast (11.63 ±0.26 µm; t-test p value>0.05; Fig. 3c; Appendix A Table A1) and coccosphere size (18.05 ±0.18 µm; t-test p value>0.05; Fig. 3d; Appendix A Table A1)."

- I recommend adjusting how data are presented in the text by changing specific references to the following format: 'average±SD (t-test p-value<X; Table X)'.
  **REPLY:** The data have are now presented as suggested

**Discussion**

Major comments:
Major comments:
The authors only include the impact of atmospheric $CO_2$ levels in the interpretation of the data, excluding the rest of the carbonate chemistry data presented in Table 1. Coccolithophores require $HCO_3^-$ as a substrate for calcification. The authors show a drop in pH and [$HCO_3^-$] when $CO_2$ increased from 295 to 600 µatm (Table 1), but do not include these variables when interpreting the data. This leads to an incomplete interpretation of the data. What about the lower pH? Is it possible that the variability in malformed coccoliths at 600µatm $CO_2$ is due to the combination of lower pH (unfavorable for calcite precipitation), lower $HCO_3^-$ concentration (substrate for calcification), and Omega > 1 (calcite formation still slightly favored)?

**REPLY:** We acknowledge that C-system parameters other than $CO_2$ can influence coccolithophore calcification and physiology more generally. We agree with the reviewer that the Discussion will benefit from an amendment including this topic, and have therefore added the text quoted below. We will, however, emphasize that our dataset is not suited to identify the parameter of the C-system causing potential effects. Briefly, the reason is that our C-system manipulation is a so-called DIC manipulation resulting in a coupled C-system. To identify the parameter causing adverse effects a combined TA/DIC manipulation and the resultant decoupled C-system is required (Kaczmarek et al., 2015; Keul et al., 2013; Langer and Bode, 2011). Another important point is that our C-system manipulation did not cause any changes in morphology, morphometry, and PIC/POC. For that reason alone, identifying a parameter causing changes is not possible. As for the potential effect of lower DIC at the 600 µatm $CO_2$ treatment, please see our reply above (in the comments to the Methods section).

**NEW TEXT:** see reply to comment on Line 23-25

Some line-by-line comments:

Line 254-255: This statement is not true. See comments on Results for details.
**REPLY:** Correct.

**The text now reads:**
"In this work, for the first time, we show that the percentage of malformed coccoliths in H. carteri does not change in a significant way moving from 295 to 600 µatm $CO_2$."

Line 266-274: I don't understand the point of Lines 266-270.**REPLY:** We clarified this point. The text formerly in Line 266-277 now reads:

**NEW TEXT:** "The comparison of malformations in different strains/species at one single $CO_2$ level is instructive but not sufficient to assess C-system effects. Malformations in coccolithophores vary both between strains/species and over time in a single strain under constant environmental conditions (Langer et al., 2009, 2013; Langer and Benner, 2009). A better assessment of C-system effects on coccolithophores is achieved when comparing trends of different experiments rather than absolute values of different experiments (Hoppe et al., 2011). Such a comparison clearly suggests species specific responses to $CO_2$, identifying more/less sensitive species".

Line 272: "...the negative effect of carbonate chemistry". This phrase does not make sense.
**REPLY:** The phrasing has been changed (see reply to previous comment).

Line 275-276: "the effect of proton inhibition" The authors do not present any data on protons (e.g., pH) from the previous work cited.
**REPLY:** The phrasing has been changed (see reply to previous comment).

Line 278-279: "less sensitive to acidification" The authors did not include any carbonate chemistry data (aside from atmospheric $CO_2$) when referencing coccolith malformation

documented throughout the literature. It is inappropriate to make a claim about sensitivity to acidification without showing the relevant acidification data.

**REPLY:** We clarified this point. See reply to the comment on the abstract.

Figure 4: What about the pH in the other experiments? And other carbonate chemistry parameters (i.e., $HCO_3^-$, $CO_3^{2-}$, DIC, etc...)? It is difficult to interpret comparisons when only $CO_2$ µatm is included since the carbonate chemistry of the growth medium can varying based on the buffering capacity in seawater.

**REPLY:** We now discuss the issues related to different C-system parameters. See reply to the comment on the abstract.

Figure 4b-c: Is this just showing the data from panel a) again?

**REPLY:** Figure 4a presents the data on the percentages of malformed and normal coccoliths in more detail (e.g., slightly malformed, malformed, fragmented), while Figures 4b and c show the overall difference in the percentages of malformed coccoliths in a simpler way.

Figure 5: What were the other relevant conditions in the other study, aside from pCO$_2$?

**REPLY:** The other study also used a coupled C-system so that results are directly comparable to our study (see also reply to the comment on the abstract).

The rest of the Discussion focuses on comparing the findings in the manuscript to previous work. This section will need to be revised accordingly after the issues identified throughout the manuscript are resolved.

**REPLY:** We resolved the issues and revised the Discussion (see above).

**Note for the editor:**

While checking the manuscript we noticed an error in the Section 4.1 (Malformations in H. carteri in response to $CO_2$ increase) related to the strain B92/11 of E. huxleyi. In the figure, and consequently in the discussion, the wrong data for E. huxleyi B92/11 had been included. The data in Figure 4a, b, c have therefore been corrected, along with the corresponding text.

The new text has been modified to fix this error as well as it takes into account the comments from reviewer 2.

**We have fixed the text and figure as follows:**

Lines 261-279: **OLD TEXT:** "When considering responses to $CO_2$ levels close to 600 µatm, the percentages of malformed coccoliths in *E. huxleyi* (RCC1238 and RCC1256) are lower than *H. carteri* (Fig. 4a, c). In contrast, the heavily calcified species *C. leptoporus* (RCC1168) and *C. quadriperforatus* (RCC1141) consistently show a higher percentage of malformed coccoliths compared to *H. carteri* (~90%, Fig. 4a, c). It is interesting to note that these percentages also include a significant amount of extremely malformed coccoliths (89% for C. quadriperforatus RCC1141, and 46% for *C. leptoporus* RCC1168; Fig. 4a). This degree of malformation has never been observed in our

experiments. Since biological parameters such as coccolith morphology undergo numerical changes over time (Langer et al., 2013), the assessment of species sensitivity should not be based on the morphology of different strains or species at one $CO_2$ level alone, but rather the change of morphology in response to a change in $CO_2$ (Hoppe et al., 2011). For example, the species *E. huxleyi* strain B92/11 shows varying percentages of malformations across a narrow range of $CO_2$ levels, illustrating this observation (Fig. 4a, b). On the other hand, at ca. 600 µatm $CO_2$ *Calcidiscus* displays more malformations than at 295 µatm of $CO_2$ (Fig. 4c), and the reason for this is most likely that the negative effect of carbonate chemistry is almost invisible at ca. 300 µatm $CO_2$. Using the reasoning of Bach et al. (2015), we would say that at 300 µatm of $CO_2$ neither substrate limitation nor proton inhibition play a significant role, and the malformations depend on other experimental conditions.

[revised manuscript text omitted]

**Rebuttal letter Reviewer 2**

The researchers Sthephania Bianco and colleagues evaluate the response of the coccolithophore species Helicosphaera carteri to changes in $CO_2$ concentrations using laboratory manipulation experiments. A single strain was cultivated under pre-industrial $CO_2$ levels according to the IPCC SSP 2-4.5 scenario for 2100. Results reveal limited changes in POC and PIC production, as well as, in coccolith morphology, under future $CO_2$ levels. Based on these results, authors conclude that this species will most likely not experience substantial changes in its performance under a high-$CO_2$ scenario expected by the end of the century.

A large body of evidence indicates that projected changes in marine carbonate chemistry driven by human activities will be detrimental for coccolithophore perfomance. Given their abundance and fundamental role in the biological and carbonate pumps, changes in coccolithophore abundance, composition and/or degree of calcification will most likely have impacts on the oceanic carbon cycle and marine ecosystems. Therefore, there is an urgent need for studies like this one to evaluate how changes in $CO_2$ concentration will affect keystone marine organisms and ecosystems. This is particularly important for large coccolithophore species, which, despite their relatively low abundance, play a major role in the carbon cycle. Given the limited existing information about the response of H. carteri to $CO_2$ changes, I consider the information provided in this paper valuable and worth publishing.

Overall, the manuscript is reasonably well written, the results are valuable, the figures are appropriate, and the interpretation of the data will be useful for the scientific community. Therefore, I recommend acceptance of this manuscript after the comments listed below are implemented (moderate revision).
**REPLY:** We appreciate the reviewer's positive evaluation of our manuscript.

Specific comments

Line 29 "…unaltered general health". Please rephrase this sentence avoiding the use of the term health.
**REPLY: The sentence now reads:** "… unaltered physiological state".

Line 40 "…different species is required".

As mentioned later on in the manuscript, even different responses within strains and varieties of the same species have been documented. So please, include this nuance here in the introduction.
**REPLY:** we have added this nuance. **The sentence now reads:**

"…different species, and even strains (Langer et al., 2009), is required".

Lines 55-57. This data are valuable but somewhat missleading. For *E. huxleyi* values are provided in pg of Calcite (i.e. $CaCO_3$) while for *H. carteri* the units are different (pg of

Carbon). Could authors provide the data in the same units to facilitate comparison between both species? this would facilitate direct comparison between species.
**REPLY:** we converted all units to [pg C].

Lines 57. Including a general description of the ecology of this species here (or somewhere in the introduction) together with a description of the geographical distribution of this species would be helpful. This information would allow the reader to better understand the relevance of this species on a global context.
**REPLY: We have added the following:**

"*Helicosphaera carteri* is generally considered to be a species typical of warm waters (e.g., Baumann et al., 2005; Brand, 1994), with moderately high-nutrient levels (e.g., Andruleit and Rogalla, 2002; Findlay and Giraudeau, 2000, 2002; Ziveri et al., 1995, 2004). However, it has a general wide distribution (as reported in the CASCADE database; de Vries et al., 2024) and it seems to be an opportunistic species, easily adaptable to different environmental conditions (Dimiza et al., 2015 and references therein). This adaptability of *H. carteri* is confirmed by its long fossil record, spanning back more than 20 Million years (Aubry, 1988; Young, 1998)."

Line 80. Please provide information about the locality from which strain RCC1323 was retrieved.
**REPLY:** The information was added.

Line 144. Could authors provide the magnification used for the analysis? What is the error of these measurements?
**REPLY:** The magnification and the error are added.

**The text has been changed as follows:**

**Lines 143-146:** Coccosphere size ($\varnothing$), aspect ratio ($AR_{coccosphere}$) and roundness ($RD_{coccosphere}$) data were obtained by photographing more than 50 coccospheres per each replicate using an inverted microscope Leica CMS-D35578 at 400x magnification and a Leica Camera Ltd CH-9435. The images were processed with ImageJ software (Rueden et al., 2017; Appendix A Fig. A1) using a customized macro (https://github.com/mbordiga/Coccoliths).

The estimated standard error of the mean are 0.1219 for $\varnothing$, 0.006119 for $AR_{coccosphere}$ and 0.004549 for $RD_{coccosphere}$ at 295 µatm; while at 600µatm are: for 0.1233 $\varnothing$, 0.006399 for $AR_{coccosphere}$ and 0.004781 for $RD_{coccosphere}$."

Line 201. This sentence could be clearer if authors provide % of malformed coccoliths for both treatments. It is a bit confusing to use the percentage of malformed coccoliths in the first part of the sentence and the normal coccoliths in the second part.
**REPLY: We have adjusted the sentence as follows and following the indications of Reviewer 1. The sentence now reads:**

"…in the second treatment the malformed coccoliths account for an average of 10.65±10.82% (Table 2, Fig. 2)."

Lines 208-209. In the previous paragraph you used the past tense and the present in this one. Please revise the verb tense.
**REPLY: Done.**

Lines 220 – 222. Please include information about the changes in POC and PIC in the experiment in the abstract.
**REPLY: We added the following to the abstract:**

**NEW TEXT: Lines 27-29:** "Our results indicate that *H. carteri* morphology is not significantly affected by increasing $CO_2$, in contrast to other heavily calcified species. *Helicosphaera carteri* protoplast and coccosphere shapes did not vary with changes in $CO_2$, as did its particulate inorganic carbon (PIC) and particulate organic carbon (POC) quotas, as well as the PIC:POC ratio, indicating unaltered general physiological state."

Line 243. Müller et al. (2015) could be cited here as well.
**REPLY: citation added.**

Line 265. Some authors consider this species a sub-species (https://www.mikrotax.org/Nannotax3/index.php?taxon=Calcidiscus%20leptoporus%20subsp.%20quadriperforatus&module=ntax_cenozoic). Please, clarify this point in the text.

**REPLY:** RCC1141 is *C. leptoporus* according to the RCC website (https://roscoff-culture-collection.org/rcc-strain-details/1141). We corrected this mistake.

**We added the following:**

**NEW TEXT:** "Today *C. leptoporus* and *C. quadriperforatus* are mostly considered separate species (https://roscoff-culture-collection.org/rcc-strain-details/1141), although some authors prefer to consider *quadriperforatus* a sub-species (https://www.mikrotax.org/; Young et al., 2022). For a detailed discussion of the taxonomical status of *Calcidiscus* see Geisen et al. (2004)."

Lines 272-274. This sentence is not clear enough. Somewhere in the text, authors should provide background information explaining the underlying reasons of the production of malformed coccoliths. As it is written now, this sentence assumes the reader already know the mechanisms behind this change.
**REPLY: We clarified this point. Reviewer 1 also criticised this part. We have restructured the argument to make the point clear. The text formerly in Line 266-277 now reads:**

**NEW TEXT:** "The comparison of malformations in different strains/species at one single $CO_2$ level is instructive but not sufficient to assess C-system effects. Malformations in coccolithophores vary both between strains/species and over time in a single strain under constant environmental conditions (Langer et al., 2009, 2013; Langer and Benner, 2009). A better assessment of C-system effects on coccolithophores is achieved when comparing trends of different experiments rather than absolute values of different experiments (Hoppe et al., 2011). Such a comparison clearly suggests species specific responses to $CO_2$, identifying more/less sensitive species".

Figure 4. Could authors include a legend with the names of the species on the side or at least indicate between brackets the colour used for the different species?. The non-specialized reader won't be able to differentiate the species based on their coccosphere. **REPLY:** Good point. We added a colour code legend.

**The figure has been adjusted as follows:**

[Figure]

Line 308. The symbol used for the diameter is different from the one used in the first sentence of this paragraph of this section.
**REPLY:** We now use only one symbol for the diameter.

Line 321. Could authors explain better this sentence "…the obligate calcifier-nature of the genus Heliscosphaera"

**REPLY: We added the following after the sentence quoted by the reviewer:**

**NEW TEXT:** "Some coccolithophores such as *Coccolithus braarudii* are obligate calcifiers, i.e. they need to calcify, whereas others such as *Emiliania huxleyi* are facultative calcifiers, i.e. they do not necessarily need to calcify (Walker et al., 2018). As per our own observation, and the extensive observational record available at the RCC Roscoff (https://roscoff-culture-collection.org/; I. Probert, personal communication) *H. carteri* is an obligate calcifier which might imply a stable PIC/POC ratio because a complete coccosphere is essential for survival (Šupraha & Henderiks, 2020; Walker et al., 2018)"

Line 332. Can authors provide an explanation for the different sensitivity of coccolithophore species based on the PI:POC ratio?
**REPLY:** We added the following explanation.

**"OLD TEXT...** and Umbilicosphaera sibogae (0.62; Gafar et al., 2019b).

**NEW TEXT:** "The latter authors hypothesize that a high PIC/POC ratio produces a high cellular proton load that is particularly harmful under Ocean Acidification conditions. More recently a cellular mechanism underpinning the hypothesis of Gafar et al. (2019b) was proposed (Kottmeier et al., 2022). This cellular mechanism involves Hv-type plasma-membrane proton channels which close under Ocean Acidification conditions therewith preventing proton export out of the cell with cytosolic acidification ensuing.

**OLD TEXT**: The low sensitivity of species with lower PIC:POC...".

Lines 333-334. Can authors provide examples of species with high PIC:POC ratio as well? Please also provide an explanation somewhere in the discussion about the underlying reasons of the different sensitivity to ocean acidification of coccolithophore species with low and high PIC:POC ratio.
**REPLY:** We added examples of high PIC/POC species. As for an explanation of differential sensitivities to OA see reply to previous comment.

**OLD TEXT (Line 333):** "… species with a higher PIC:POC ratio… **NEW TEXT:** such as C. leptoporus (2.08) and G. oceanica (1.25)… **OLD TEXT:** … should be more sensitive…".

- Corrected the graphic of Tables 3 and A1.

---

## Referee Report (RR1)

Dear Dr Marañón,

I've read the revised version of the manuscript "**Low sensitivity of a heavily-calcified coccolithophore under increasing CO2: the case study of *Helicosphaera carteri***" by Dr Stefania Bianco and colleagues as well as their answer to reviewer's comments. To my view and knowledge, the authors have properly addressed the reviewers' concerns and therefore, the *manuscript can be accepted as it is*.

---

## Referee Report (RR2)

**Overall**

The aim of the study is worthwhile, as it focuses on a relatively understudied coccolithophore (compared to model species such as *Emiliania/Gephyrocapsa huxleyi* and *Gephyrocapsa oceanica*). The authors investigated whether elevated $pCO_2$ impacts *Helicosphaera carteri*, assessing coccolith morphology and particulate inorganic and organic carbon (PIC and PIC, respectively). The authors claim that the results of this study suggest that *H. carteri* may have a constant contribution to the rain ratio under ocean acidification.

The revised manuscript is significantly improved over the previous version. The authors' did a good job incorporating feedback from myself and the other reviewer, which has strengthened the manuscript. Notably, they have added text to the methods and discussion clarifying their experimental setup and the lower DIC under OA conditions. I only have one major concern remaining (Fig. 2), as well as some minor revisions and recommendations for the text.

**Figure 2.**

I appreciate the authors' desire to improve the readability of the manuscript by present the data in multiple formats, but my concern is that, currently, the figure is misleading and detracts from the conclusion—that there is no difference in the percentage of malformed coccoliths under increase $CO_2$. Many (if not most) readers tend to look at figures first when reading results. It is necessary to incorporate error in some way so that readers cannot be misled by the figure. There are methods to incorporate error bars on stacked bar plots, but if that is not feasible, perhaps the type of figure should be changed.

**Minor Comments and Recommendations**

*Introduction*

Line 58: "...*H. carteri* produces between ~80 120 pg cell$^{-1}$ day$^{-1}$". Missing an 'and' or dash
Line 70: "Indeed, despite the coccosphere's function is still unclear...". The grammar is a bit unclear. Rephrase for clarity.

*Materials and Methods*

Table 1 caption: Define SD in caption.
Line 176: "$C_L^3$ is the coccolith length." Is there a typo here? Should it be $C_L$, $C_L^3$?

*Results*

Lines 224-226: I don't think it is necessary to state that the high standard deviation (SD) is due to high variability among the replicates (and vice versa for low SD and low variability). It's a bit redundant.
Lines 241-242: I recommend condensing into one sentence and rephrasing.
Line 246: Define protoplast and coccosphere sizes as 'dimensionless' in methods. Eliminate 'µm/µm'.

Lines 246-249: There only seems to be one set of values reported (600 µatm?). Is it meant to include the values at 295 µatm as well?

Line 250-251: "The range of..." This seems to repeat the previous sentence, except it also states trends that are not statistically significant (i.e., "slightly higher range"). I recommend eliminating.

*Discussion*

Line 259: Section 4.1 header may be a bit misleading as written. Consider rephrasing to clarify that elevated $CO_2$ did not lead to increased malformations.

Lines 330-331: Its awkward as written. Consider rephrasing and avoid 'good health'.

Lines 338-342: "A non-significant variation" sounds awkward. It seems like you are drawing attention to the variability, not the lack of a significant difference. Clarify that Le Guevel et al (2024) did not observe a difference in PIC:POC with changing $CO_2$, but coccosphere size was impacted.

Line 346: "global decreasing trend in $CO_2$". Consider providing the range of $CO_2$ to make it more relevant (i.e., is ~600µatm represented in this study?)

Lines 335-337: "could represent an advantage in future oceans where the species could play a stable role in the C cycle despite changes in $CO_2$ concentrations." What kind of advantage? Is it an advantage to *H. carteri*? Isn't the point that stable PIC:POC means the contribution to the rain ratio should remain stable over elevated $CO_2$ concentrations?

Lines 385-387: "The most likely explanation of these observations is that other aspects than the PIC:POC ratio influence the species' response to increased $CO_2$ levels." Isn't this reversed? Aspects other than $CO_2$ influence the PIC:POC ratio.

---

## Author Response (AR2)

**Reply to the Associate Editor**

We have carefully addressed all the points raised by the reviewers and the editor, revising our manuscript accordingly. We sincerely appreciate the reviewers' insightful comments, which have helped us improve the clarity and quality of our work.

Both Referee 1 and the Associate Editor highlighted some minor aspects related to the clarity of certain sentences and Figure 2. We have made the necessary adjustments to the text and the figure to enhance readability and ensure clarity.

**Rebuttal letter referee nr. 1**

**Overall**

The aim of the study is worthwhile, as it focuses on a relatively understudied coccolithophore (compared to model species such as *Emiliania/Gephyrocapsa huxleyi* and *Gephyrocapsa oceanica*). The authors investigated whether elevated $pCO_2$ impacts *Helicosphaera carteri*, assessing coccolith morphology and particulate inorganic and organic carbon (PIC and PIC, respectively). The authors claim that the results of this study suggest that *H. carteri* may have a constant contribution to the rain ratio under ocean acidification.

The revised manuscript is significantly improved over the previous version. The authors' did a good job incorporating feedback from myself and the other reviewer, which has strengthened the manuscript. Notably, they have added text to the methods and discussion clarifying their experimental setup and the lower DIC under OA conditions. I only have one major concern remaining (Fig. 2), as well as some minor revisions and recommendations for the text.

**REPLY:** We appreciate the reviewer's positive assessment of our revised manuscript. We have added the error to Fig. 2 and prepared a second revision incorporating the reviewer's suggestions.

**Figure 2.**

I appreciate the authors' desire to improve the readability of the manuscript by present the data in multiple formats, but my concern is that, currently, the figure is misleading and detracts from the conclusion—that there is no difference in the percentage of malformed coccoliths under increase $CO_2$. Many (if not most) readers tend to look at figures first when reading results. It is necessary to incorporate error in some way so that readers cannot be misled by the figure. There are methods to incorporate error bars on stacked bar plots, but if that is not feasible, perhaps the type of figure should be changed.

**REPLY:** We adjusted the fig. 2 following the suggestions.
**NEW FIGURE:**

[Figure]

**Figure 2.** Percentages (%) of normal and malformed coccoliths of *H. carteri*. Values reported represent the averages of the three replicates. Error bars show standard deviation.

**Minor Comments and Recommendations**

*Introduction*

- Line 58: "...*H. carteri* produces between ~80 120 pg cell$^{-1}$ day$^{-1}$". Missing an 'and' or dash
  **REPLY:** We adjusted the sentence.
  **NEW TEXT:** "..*H. carteri* produces between ~80 and ~120 pg cell$^{-1}$ day$^{-1}$".

- Line 70: "Indeed, despite the coccosphere's function is still unclear...". The grammar is a bit unclear. Rephrase for clarity.
  **REPLY:** We replaced this sentence.
  **NEW TEXT:**
  **Lines 71-72:** "Both calcite production and coccospheres are beneficial for coccolithophores in terms of eco-physiology and evolution (e.g., Henriksen et al., 2003; Langer et al., 2021; Monteiro et al., 2016; Walker et al., 2018)."

*Materials and Methods*

- Table 1 caption: Define SD in caption.
  **REPLY:** We added the SD definition in the caption.

- Line 176: "CL$^3$ is the coccolith length." Is there a typo here? Should it be $C_L$, $C_{L3}$?
  **REPLY:** The coccolith length is represented by $C_L$. We adjusted the typo.

*Results*

- Lines 224-226: I don't think it is necessary to state that the high standard deviation (SD) is due to high variability among the replicates (and vice versa for low SD and low variability). It's a bit redundant.
  **REPLY:** We have modified the sentences in order to avoid redundancy.
  **NEW TEXT:**
  **Lines 224-225:** "The percentage of malformed coccoliths at 600 µatm is characterized by a high standard deviation (SD). On the contrary, at 295 µatm, SD is quite low in all the considered categories (Table 2)."

- Lines 241-242: I recommend condensing into one sentence and rephrasing.
  **REPLY:** We have modified the text.
  **NEW TEXT:**
  **Lines 240-241:** "Cellular POC returns an average of 108.14±5.42 pg cell$^{-1}$ at 295 µatm and 118.51±6.41 pg cell$^{-1}$ at 600 µatm of $CO_2$, with an unpaired t-test showing no significant change between $CO_2$ levels (t-test p value>0.05; Table 3)."

- Line 246: Define protoplast and coccosphere sizes as 'dimensionless' in methods. Eliminate 'µm/µm'.
  **REPLY:** We eliminated 'µm/µm' and added a specific in the methods section.
  **NEW TEXT:**
  **Lines 166-168:** "Since AR and RD are based on the ratio between major and minor axes of the coccosphere and/or the protoplast, they are considered dimensionless. Hence, the unit for these parameters is not reported."

- Lines 246-249: There only seems to be one set of values reported (600 µatm?). Is it meant to include the values at 295 µatm as well?
  **REPLY:** Yes, the values included both the values from 295 and 600 ppm. We adjusted the text to make it more clear.
  **NEW TEXT:**
  **Lines 241-250:** "A non-significant change is also observed in cellular PIC and in the PIC:POC ratio, showing an average value of 151.86±4.23 pg cell$^{-1}$ at 295 µatm and 149.47±9.49 cell$^{-1}$ at 600 µatm of $CO_2$ (t-test p value>0.05; Table 3) and of 1.37±0.072 at 295 µatm and 1.27±0.013 at 600 µatm of $CO_2$, respectively (t-test p value >0.05; Table 3).

*Helicosphaera carteri* protoplast (0.90±0.02 and 0.90±0.01 at 295 µatm and 600 µatm, respectively) and coccosphere (0.89±0.02 at 295 µatm and 0.88±0.003 at 600 µatm) roundness does not show any significant variation with increasing $CO_2$ (t-test p value>0.05), indicating the maintenance of a constant shape at different $CO_2$ levels (Fig. 3a, b; Appendix A Table A1). No changes have been detected for protoplast (11.45±0.19 µm at 295 µatm and 11.81±0.27 µm at 600 µatm; t-test p value>0.05; Fig. 3c; Appendix A Table A1) and coccosphere size (18.18 ±0.25 µm and 17.92±0.66 at 295 µatm and 600 µatm, respectively; t-test p value>0.05; Fig. 3d; Appendix A Table A1)."

- Line 250-251: "The range of..." This seems to repeat the previous sentence, except it also states trends that are not statistically significant (i.e., "slightly higher range"). I recommend eliminating.
  **REPLY:** We appreciate the suggestion. We eliminated the repetition.

*Discussion*

- Line 259: Section 4.1 header may be a bit misleading as written. Consider rephrasing to clarify that elevated $CO_2$ did not lead to increased malformations.
  **REPLY:** We replaced "malformations" by "morphology" (**Line 258**).

- Lines 330-331: It's awkward as written. Consider rephrasing and avoid 'good health'.
  **REPLY:** We changed the sentence.
  **NEW TEXT:**
  **Lines 329-330:** "Daily observation of the living culture under a light microscope showed in both $CO_2$ treatments that *H. carteri* remained in a good condition, with good motility of the cells."

- Lines 338-342: "A non-significant variation" sounds awkward. It seems like you are drawing attention to the variability, not the lack of a significant difference. Clarify that Le Guevel et al (2024) did not observe a difference in PIC:POC with changing $CO_2$, but coccosphere size was impacted.
  **REPLY:** We adjusted the text.
  **NEW TEXT:**
  **Lines 336-338:** "The maintenance of a stable PIC:POC ratio in the same *H. carteri* strain and at similar $CO_2$ levels (300 µatm and 600 µatm) has recently been observed also by Le Guevel et al. (2024) (Fig. 5), who also recorded a slight increase in coccosphere size within this $CO_2$ range (+0.69 µm from 200 to 600 µatm). These authors grew the species under even higher $CO_2$ levels, recording a decrease in coccosphere size (-1.05 µm) moving from 600 µatm to 1400 µatm of $CO_2$. However, this decrease in coccosphere size with

increasing $CO_2$/decreasing pH was not associated with a significant trend in the PIC:POC ratio (Le Guevel et al., 2024).”

- Line 346: “global decreasing trend in $CO_2$”. Consider providing the range of $CO_2$ to make it more relevant (i.e., is ~600µatm represented in this study?)
  **REPLY:** In the cited works, the $CO_2$ range considered spans from the warm, high-$CO_2$ world of the Middle Miocene to the cooler, low-$CO_2$ conditions of the Pleistocene. Specifically, $CO_2$ levels ranged from 350–500 ppm during the Middle Miocene to as low as 200 ppm in the Pleistocene.

  We adjusted the text by adding the range of $CO_2$.
  **NEW TEXT:**
  **Lines 345-348:** “These authors documented a stable PIC:POC ratio of this genus along with a reduction of coccolith (and coccosphere) size in response to the global decreasing trend in $CO_2$, which ranged from ~350–500 ppm during the Middle Miocene to ~200 ppm in the Pleistocene (Herbert et al., 2016; Sosdian et al., 2018; Super et al., 2018; Zachos et al., 2001; Zhang et al., 2013).”

- Lines 335-337: “could represent an advantage in future oceans where the species could play a stable role in the C cycle despite changes in $CO_2$ concentrations.” What kind of advantage? Is it an advantage to *H. carteri*? Isn't the point that stable PIC:POC means the contribution to the rain ratio should remain stable over elevated $CO_2$ concentrations?
  **REPLY:** We clarified this by rephrasing as follows:
  **NEW TEXT:**
  **Lines 357-358:** “…could stabilize the future C-cycle despite changes in $CO_2$ concentrations.”

- Lines 385-387: “The most likely explanation of these observations is that other aspects than the PIC:POC ratio influence the species' response to increased $CO_2$ levels.” Isn't this reversed? Aspects other than $CO_2$ influence the PIC:POC ratio.
  **REPLY:** No, it is not reversed. However, we changed the phrasing to make it clearer.
  **NEW TEXT:**
  **Lines 386-388:** “The most likely explanation for these observations is that the PIC:POC ratio is not a sufficient predictor for the strain's sensitivity to increased $CO_2$."

**References**

- Herbert, T. D., Lawrence, K. T., Tzanova, A., Peterson, L. C., Caballero-Gill, R., and Kelly, C. S.: Late Miocene global cooling and the rise of modern ecosystems, Nature Geosci, 9, 843–847, https://doi.org/10.1038/ngeo2813, 2016.

- Henriksen, K., Stipp, S. L. S., Young, J. R., and Bown, P. R.: Tailoring calcite: Nanoscale AFM of coccolith biocrystals, American Mineralogist, 88, 2040–2044, https://doi.org/10.2138/am-2003-11-1248, 2003.

- Langer, G., Taylor, A. R., Walker, C. E., Meyer, E. M., Ben Joseph, O., Gal, A., Harper, G. M., Probert, I., Brownlee, C., and Wheeler, G. L.: Role of silicon in the development of complex crystal shapes in coccolithophores, New Phytologist, 231, 1845–1857, https://doi.org/10.1111/nph.17230, 2021.

- Le Guevel, G., Minoletti, F., Geisen, C., Duong, G., Rojas, V., and Hermoso, M.: Multispecies expression of coccolithophore vital effects with changing $CO_2$ concentrations and pH in the laboratory with insights for reconstructing $CO_2$ levels in geological history, https://doi.org/10.5194/egusphere-2024-1890, 28 June 2024.

- Monteiro, F. M., Bach, L. T., Brownlee, C., Bown, P., Rickaby, R. E. M., Poulton, A. J., Tyrrell, T., Beaufort, L., Dutkiewicz, S., Gibbs, S., Gutowska, M. A., Lee, R., Riebesell, U., Young, J., and Ridgwell, A.: Why marine phytoplankton calcify, Sci. Adv., 2, e1501822, https://doi.org/10.1126/sciadv.1501822, 2016.

- Sosdian, S. M., Greenop, R., Hain, M. P., Foster, G. L., Pearson, P. N., and Lear, C. H.: Constraining the evolution of Neogene Ocean carbonate chemistry using the boron isotope pH proxy, Earth and Planetary Science Letters, 498, 362–376, https://doi.org/10.1016/j.epsl.2018.06.017, 2018.

- Super, J. R., Thomas, E., Pagani, M., Huber, M., O'Brien, C., and Hull, P. M.: North Atlantic temperature and pCO2 coupling in the early-middle Miocene, Geology, 46, 519–522, https://doi.org/10.1130/G40228.1, 2018.

- Walker, C. E., Taylor, A. R., Langer, G., Durak, G. M., Heath, S., Probert, I., Tyrrell, T., Brownlee, C., and Wheeler, G. L.: The requirement for calcification differs between ecologically important coccolithophore species, New Phytologist, 220, 147–162, https://doi.org/10.1111/nph.15272, 2018.

- Zachos, J., Pagani, M., Sloan, L., Thomas, E., and Billups, K.: Trends, Rhythms, and Aberrations in Global Climate 65 Ma to Present, Science, 292, 686–693, https://doi.org/10.1126/science.1059412, 2001.

- Zhang, Y. G., Pagani, M., Liu, Z., Bohaty, S. M., and DeConto, R.: A 40-million-year history of atmospheric CO2, Phil. Trans. R. Soc. A., 371, 20130096, https://doi.org/10.1098/rsta.2013.0096, 2013.